# Learning Discriminative and Generalizable Anomaly Detector for Dynamic Graph

## Abstract

Anomaly detection in dynamic graphs is critical for many real-world applications but remains challenging because labeled anomalies are scarce. Most existing approaches rely on unsupervised or semi-supervised learning, which often struggle to learn discriminative representations and generalize to unseen cases. To overcome these issues, we propose SDGAD, a supervised framework with three main components. First, we design a residual representation that highlights deviations from historical patterns, providing strong anomaly signals. Second, we constrain the residuals of normal samples within an interval defined by two co-centered hyperspheres, ensuring consistent scales while keeping anomalies separable. Third, we use a normalizing flow to model the likelihood distribution of normal samples, treating anomalies as out-of-distribution points. Based on this distribution, we derive an explicit decision boundary and further propose a bi-boundary optimization strategy to boost generalization. Experiments on six datasets, covering both real and synthetic anomalies, show that SDGAD consistently outperforms diverse baselines across multiple evaluation metrics. The code is available at this repository:https://anonymous.4open.science/r/SODA-7EFD/.

## 1 Introduction

Dynamic graph anomaly detection (DGAD) is vital for real-world applications such as detecting financial fraud (Zhang et al., 2021; Li et al., 2023), abnormal social interactions (Berger-Wolf & Saia, 2006; Greene et al., 2010), cyberattacks (Zhang et al., 2022), and has therefore attracted increasing research attention. Previous DGAD methods (Yu et al., 2018; Zheng et al., 2019; Cai et al., 2021; Liu et al., 2023) rely on discrete-time dynamic graph (DTDG) representations, which capture temporal evolution at the snapshot level but inevitably lose fine-grained temporal information. Recent studies (Tian et al., 2023; Postuvan et al., 2024; Yang et al., 2024) employ continuous-time dynamic graph (CTDG) representations to alleviate this issue. Nonetheless, both DTDG- and CTDG-based DGAD methods still face a fundamental challenge: anomalies in real-world scenarios are far rarer than normal instances, resulting in severe class imbalance.

To cope with the scarcity of labeled anomalies, most existing methods (Yu et al., 2018; Cai et al., 2021; Liu et al., 2023; Postuvan et al., 2024; Yang et al., 2024) adopt unsupervised learning setting, where models are trained only on normal or unlabeled data and then flag anomalies as deviations from learned normal patterns. However, without explicit supervision, the resulting decision boundaries are often ambiguous, leading to poor discriminability. As illustrated in Figure 1 (a), anomaly scores can collapse into a narrow low-valued range in which normal and anomalous samples are largely indistinguishable. This limitation is particularly problematic in high-stakes domains such as financial fraud detection, where the goal is to make precise binary decisions on individual samples rather than to generate a ranking over a set of candidates. Semi-supervised approaches (Zheng et al., 2019; Tian et al., 2023) attempt to leverage the few available labeled anomalies and augment them with pseudo-labeled data. Yet pseudo labels are inherently noisy, and the observed available anomalies usually cover only a limited subset of anomaly types. Consequently, these methods are prone to overfitting the seen patterns, introducing bias and limiting generalization to unseen anomalies.

The above observations highlight a critical need for a DGAD method with stronger discriminability and generalizability. Achieving this objective requires two essential capabilities: First, it must be capable of learning informative representations that provide anomaly-relevant signals with sufficiently

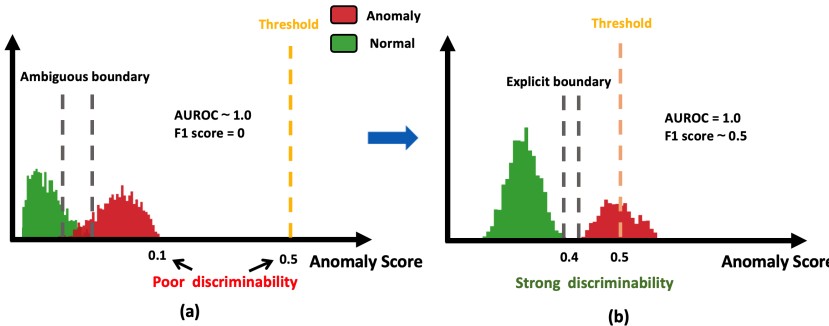

Figure 1: Conceptual illustration. (a) Unsupervised methods often yield ambiguous decision boundaries, with anomaly scores compressed into a narrow range. (b) The objective of our method.

discriminability to distinguish anomalous behaviors. Second, it must be able to construct an explicit and robust decision boundary. To this end, we propose an effective and generalizable Supervised framework for DGAD (**SDGAD**). To learn informative representations, we introduce **residual representation**, which captures the discrepancy between node's embeddings computed with and without the current interaction. Since anomalous behaviors in dynamic graphs often deviate from historical patterns, the residual representation explicitly emphasizes such deviations, providing a principled signal for detection. However, different patterns of anomalies may yield residuals of varying scales. To improve discriminability, we design a **representation restriction** mechanism that constrains the residuals of normal samples within an interval bounded by two co-centered hyperspheres, while keeping anomalies outside. Finally, we employ a normalizing flow to model the log-likelihood distribution of normal samples and identify anomalies as out-of-distribution instances. Building on this distribution, we introduce a **bi-boundary optimization** strategy to construct explicit and robust decision boundary. Our contributions are as follows:

- We introduce SDGAD, an effective and generalizable supervised framework for dynamic graph anomaly detection.

- We propose a residual representation with restriction to learn discriminative representations and a bi-boundary optimization strategy to construct explicit and robust decision boundary.

- We conduct extensive experiments on six datasets, comprising both datasets with real anomalies and datasets with synthetic anomalies. Comprehensive evaluations demonstrate that our framework achieves superior performance compared to diverse baselines.

## 2 RELATED WORK

### 2.1 ANOMALY DETECTION IN DYNAMIC GRAPHS

From a modeling perspective, existing dynamic graph anomaly detection approaches can be broadly categorized into two types: DTDG-based and CTDG-based. DTDG-based approaches (Yu et al., 2018; Zheng et al., 2019; Cai et al., 2021; Liu et al., 2023) treat dynamic graphs as sequences of snapshots, where each snapshot is considered a static graph. However, due to fixed-interval representation and coarse temporal granularity, these methods often lose critical temporal information and fail to capture the continuous spatio-temporal dependencies essential for accurate anomaly detection. To overcome these limitations, recent work has explored CTDG-based models (Postuvan et al., 2024; Reha et al., 2023; Tian et al., 2023; Yang et al., 2024) which represent interactions as event streams with continuous timestamps. Although CTDG alleviates the drawbacks of DTDG by offering finer temporal resolution, the fundamental challenges of DGAD remain unresolved. In particular, most approaches, whether DTDG- or CTDG-based, employ unsupervised learning frameworks that train exclusively on normal samples. This paradigm yields ambiguous decision boundaries, as the models lack explicit contrast between normal and anomalous patterns. Thus, some semi-supervised approaches (Zheng et al., 2019; Tian et al., 2023) attempt to leverage the few available anomaly labels by combining them with pseudo-labeled data. However, pseudo labels in-

evitably introduce noise and the scarce anomaly labels typically reflect only a narrow set of patterns, causing semi-supervised methods to overfit and limiting their generalization to unseen anomalies.

## 2.2 CLASS IMBALANCE

The extreme rarity of anomalies in real-world scenarios makes class imbalance a fundamental challenge in dynamic graph anomaly detection. Re-sampling and re-weighting strategies (Wang et al., 2019; Cui et al., 2020; Dou et al., 2020; Liu et al., 2020), though effective on static graphs, are less applicable in dynamic settings due to temporal dependencies and evolving relationships. Graph data augmentation (Hou et al., 2022; Kong et al., 2020; Chen et al., 2024; Zhao et al., 2021) has been explored as an alternative, yet most techniques are designed for static graphs, rely on node or edge attributes that are often sparse or unavailable in dynamic graphs. Although recent augmentation methods for dynamic graphs have been proposed (Tian et al., 2024a; Wang et al., 2021c), they rely on empirical heuristics without guarantees that the generated anomalous samples faithfully capture real anomaly characteristics. Consequently, enabling effective learning under such severe imbalance remains a major open challenge in dynamic graph anomaly detection.

## 3 PRELIMINARIES

**Notations.** A dynamic graph is a time-dependent graph $G = (V(t), E(t))$, where $V(t)$ and $E(t)$ denote the node and edge sets at timestamp $t$, respectively. In this paper, we adopt the CTDG formulation, which represents the dynamic graph as an ordered stream of events $G = \{\xi(t_0) \ldots \xi(t_k) \ldots \xi(t_n)\}_{k=0}^{n}$ with nondecreasing timestamps $t_0 \leq t_k \leq t_n$. Each event $\xi(t_k) = (v_i, v_j, t_k, e_{i,j}^{t_k})$ denotes an interaction between nodes $v_i$ and $v_j$ at timestamp $t_k$ with associated edge feature $e_{i,j}^{t_k}$. If the dynamic graph is non-attributed, we simply set the node and edge feature to zero vectors. Multiple interactions may occur either between the same node pair at different times or among different pairs at the same time. Each interaction is further annotated with a binary label $y \in \{0, 1\}$, where 0 indicates normal and 1 anomalous. In practice, anomaly labels are highly imbalanced, where the number of normal samples vastly exceeds that of anomalies.

**Problem definition.** Given an event $\xi(t)$ and historical events before $t$, the goal is to design a model that learns the representation for $\xi(t)$ and assigns it a continuous anomaly score in $[0, 1]$, thereby quantifying its degree of abnormality and determining whether the event is anomalous.

**Normalizing Flow** provides exact density estimation by mapping an intractable data distribution $\mathcal{Q}$ to a tractable latent distribution $\mathcal{Z}$ through an invertible transformation, which is implemented as a composition of $F$ invertible mappings: $\boldsymbol{\Phi}_\theta = \boldsymbol{\Phi}_F \circ \cdots \circ \boldsymbol{\Phi}_1$, where $\theta$ denotes the trainable parameters. For an input $x \in \mathcal{Q}$, its log-likelihood $\log[p(x)]$ can be computed as follow:

$$\log[p(x)] = \log p_{\mathcal{Z}}(\boldsymbol{\Phi}_\theta(x)) + \sum_{f=1}^{F} \log \left| \det J_{\boldsymbol{\Phi}_f}(x_{f-1}) \right| \tag{1}$$

where $J_{\boldsymbol{\Phi}_f}(x_{f-1}) = \frac{\partial \boldsymbol{\Phi}_f(x_{f-1})}{\partial x_{f-1}}$ is the Jacobian matrix, $\det$ denotes the determinant. In practice, the latent distribution $\mathcal{Z}$ is typically assumed as a standard Gaussian. Thus, the parameters $\theta$ can be optimized by maximizing the log-likelihoods across the training distribution $\mathcal{Q}$. The loss can be formulated as a maximum likelihood loss:

$$\mathcal{L}_{\mathcal{ML}} = \mathbb{E}_{x \sim \mathcal{Q}} \left[ \frac{d}{2} \log(2\pi) + \frac{1}{2} \boldsymbol{\Phi}_\theta(x)^T \boldsymbol{\Phi}_\theta(x) - \sum_{f=1}^{F} \log \left| \det J_{\boldsymbol{\Phi}_f}(x_{f-1}) \right| \right] \tag{2}$$

## 4 METHODOLOGY

Figure 2 provides an overview of the proposed framework, which comprises three main components: residual representation encoding, representation restriction, and bi-boundary optimization. They are described in detail in the following subsections.

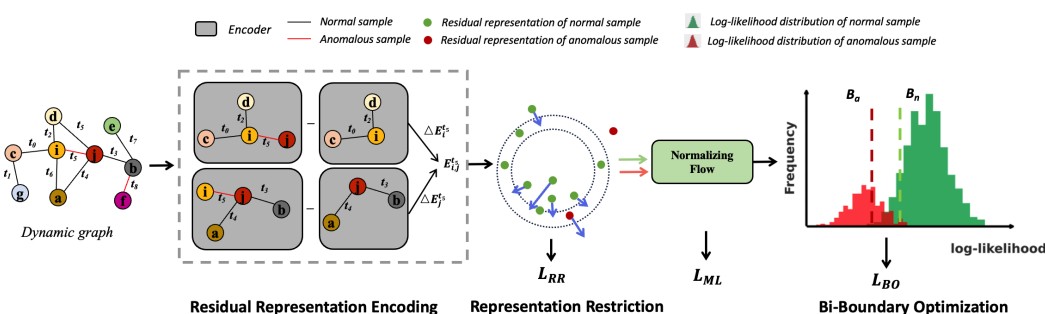

Figure 2: Overview of our proposed framework. We first encode each sample's residual representation by contrasting two node-pair embeddings. Then the residual representations of normal samples (green dots) are restricted into an interval region between two co-centered hypersphere, while anomalous samples (red dots) are pushed outside. Finally, we use the normalizing flow to model the log-likelihood distribution of normal samples and then the bi-boundary optimization is used to learn an explicit and robust decision boundary.

## 4.1 RESIDUAL REPRESENTATION ENCODING

Instead of introducing a new encoder, our framework is designed to be compatible with the encoder of any CTDG model. Since encoding mechanisms differ across models, we present a unified abstraction here, with model-specific implementations of the encoder $enc(\cdot)$ detailed in Appendix B. Formally, given an interaction $\xi(t) = (v_i, v_j, t, e_{i,j}^t)$ [1], the embedding of a target node at timestamp $t$ is computed by combining its historical interactions with the current one. Specifically, for node $v_i$, we sample its $L$ historical neighbors together with their associated edges and timestamps and append the current interaction $(v_j, e_{i,j}, t)$. The resulting inputs are then fed into the encoder:

$$\boldsymbol{E}_i^t = enc(\{v_l, e_{i,l}^t, t_l\}_{l=1}^L \| \{v_j, e_{i,j}^t, t\}) \quad v_l \in \mathcal{N}_{<t}(v_i) \tag{3}$$

where $\boldsymbol{E}_i^t$ denotes the embedding of node $v_i$ at timestamp $t$. $\mathcal{N}_{<t}(v_i)$ is the set of sampled historical neighbors of $v_i$ prior to $t$. The embedding of node $v_j$ is computed analogously.

In dynamic graphs, anomalies typically manifest as deviations from recent temporal or structural patterns. Thus, effective detection should emphasize relational signals that capture such deviations. Recall that the node embedding $\boldsymbol{E}_i^t$ is obtained by aggregating both historical and current interactions. While this aggregation captures long-term dependencies, it may obscure short-term irregularities that are often most indicative of anomalies. To detect the irregularities, we introduce the residual representation, defined as:

$$\Delta E_i^t = E_i^t - E_i^{t^-} = enc(\{v_l, e_{i,l}^t, t_l\}_{l=1}^L \| \{v_j, e_{i,j}^t, t\}) - enc(\{v_l, e_{i,l}^t, t_l\}_{l=1}^L) \tag{4}$$

This residual representation $\Delta E_i^t$ measures the discrepancy between the node embedding computed with and without the current interaction. Its norm remains small when the current interaction conforms to historical patterns, but increases when deviations occur. This property aligns with neighborhood-based principle in graph analysis: local consistency yields low representation differences, whereas anomalies induce larger ones. By suppressing stable components and emphasizing novel temporal or structural variations, the residual representation serves as a discriminative signal for anomaly detection. Then, the representation of the event $\xi(t)$ can be obtained by concatenating the residual representations of the two nodes: $E_{i,j}^t = \Delta E_i^t \| \Delta E_j^t$.

## 4.2 REPRESENTATION RESTRICTION

Although the residual representations provide informative anomaly-relevant signal, its effectiveness is limited by the diversity of anomaly patterns. Different anomalies induce residuals of varying scales: some lead to large residuals (e.g., interaction bursts), whereas others manifest only as small residuals (e.g., mild timing irregularities). This scale variation makes it challenging to find a unified

---

[1]For notational clarity, we omit the subscript $k$ in all subsequent formulations

decision boundary to consistently distinguish normal samples from anomalies: a threshold suitable for large residuals may miss subtle anomalies, while one tuned for smaller values may increase false positives. To address this, we draw inspiration from Zhang et al. (2024) and propose a representation restriction strategy to constrain the residual representations of normal samples into an interval region between two co-centered hyperspheres, while keeping anomalies outside. Specifically, we first project the residual representation $E_{i,j}^t$ through a linear layer to obtain projected representation $E_{i,j}^{t'}$. The restriction loss is then defined as:

$$\mathcal{L}_{\mathcal{RR}} = (1-y) \underbrace{\left( \max(0, A - r_{\max}) + \max(0, r_{\min} - A) \right)}_{\text{interval penalty}}$$
$$+ y \underbrace{\left( \max(A - r_{max}) + MSE(E_{i,j}^{t'}, E_{i,j}^t) + 1 - cos(E_{i,j}^{t'}, E_{i,j}^t) \right)}_{\text{discrimination + consistency}} \quad (5)$$

where $A = \sqrt{\|E_{i,j}^{t'}\|_2 + 1} - 1$ is a stabilized approximation of the $L_2$ norm. $r_{\min}, r_{\max}$ represent the radii of the inner and outer co-centered hyperspheres that bound the normal region, respectively. $MSE(\cdot)$ and $cos(\cdot)$ denote the mean square error function and cosine similarity function, respectively. The first term, referred to as the *interval penalty*, is applied to normal samples and penalizes cases where $A$ falls outside the interval $[r_{\min}, r_{\max}]$. This constraint encourages the projected representation of normal samples to remain within a compact region of consistent scale, thereby promoting the formation of a unified decision boundary. However, relying on this constraint alone risks collapsing both normal and anomalous projected representation into the same region, reducing their discriminability. Therefore, the second term is applied to anomalous samples and integrates two objectives: (i) it encourages the projected representation of anomalous samples to remain close to their initial residuals, rather than being mapped into the normal region. (ii) it enforces separation by penalizing anomalies that fall within the interval region. Together, these terms compact the residuals of normal samples while keeping anomalies separable in the latent space. This design provides the foundation for establishing an explicit decision boundary and supports the model's ability to generalize to unseen anomaly types.

## 4.3 BI-BOUNDARY OPTIMIZATION

Building on the restricted representations, we can use a distribution estimator that models the distribution of normal samples and treats anomalies as out-of-distribution deviations. Specifically, we employ normalizing flow to compute the exact log-likelihood $\log[p(x)]$ for each sample, as defined in Equation 1. Since log-likelihood value can range in $(-\infty, 0]$, a normalization constant is applied to rescale them into the range $[-1, 0]$ for more stable optimization. As training is conducted in a batch-wise manner, we denote the log-likelihoods of normal and anomalous samples within a batch as $\mathcal{D}_n = \{\log[p(x_i)]\}_{i=1}^N$ and $\mathcal{D}_a = \{\log[p(x_j)]\}_{j=1}^M$, where $N$ and $M$ denote the number of normal and anomalous samples in the batch. Then $\mathcal{D}_n$ can be regarded as an approximation of the log-likelihood distribution of normal samples. As anomalies are defined relative to the normal distribution, the decision boundary $\mathcal{B}$ can be naturally constructed on $\mathcal{D}_n$.

A straightforward way is to set $\mathcal{B}$ as the $\alpha$-th percentile of sorted normal log-likelihood distribution $\mathcal{D}_n$ and identify samples with lower log-likelihoods as anomalies. However, probability density tends to spread out in high-dimensional spaces (Kirichenko et al., 2020) and the bijective nature of normalizing flow can map anomalies into the typical set of the latent space rather than the expected tail (Kumar et al., 2021). As a result, anomalies may not consistently fall into low log-likelihood regions and can even attain unexpectedly high log-likelihood value, making it difficult to establish a robust decision boundary. Therefore, we propose a bi-boundary optimization strategy. Specifically, we define a decision margin $\tau$ (e.g., $\tau = 0.1$), refining the single boundary $\mathcal{B}$ into a normal boundary $\mathcal{B}_n$ and an anomalous boundary $\mathcal{B}_a = \mathcal{B}_n - \tau$. This margin construct a buffer zone that reduces ambiguity near the boundary and improves robustness by preserving an explicit separation between normal and anomalous regions. Accordingly, the optimization objective can be formulated as:

$$\mathcal{L}_{\mathcal{BO}} = \sum_{i=1}^N |\min(softplus(\log[p(x_i)] - \mathcal{B}_n), 0)| + \sum_{j=1}^M |\max(softplus(\log[p(x_j)] - \mathcal{B}_a), 0)| \quad (6)$$

By minimizing $\mathcal{L}_{\mathcal{BO}}$, the model enforces a clear margin between the log-likelihood of normal and anomalous samples, constraining anomalous samples to $[-\infty, \mathcal{B}_a]$ and normal samples to $[\mathcal{B}_n, 0]$. Samples outside these regions are penalized in proportion to their deviation from the boundary, encouraging normal samples to concentrate in high density areas while pushing anomalies further away. Unlike hard losses that impose discontinuous penalties at the boundary, we adopt the softplus function, $\text{softplus}(x) = \log(1 + e^x)$, which scales penalties smoothly with the degree of violation.

**Overall Loss Function.**  The overall training loss of our framework is the combination of the Eq. 2, Eq. 5 and Eq. 6 as follows:

$$\mathcal{L} = \mathcal{L}_{\mathcal{ML}} + \lambda_1 \mathcal{L}_{\mathcal{BO}} + \lambda_2 \mathcal{L}_{\mathcal{RR}} \tag{7}$$

Here $\mathcal{L}_{ML}$ is the basic loss for training the normalizing flow, and it is computed only on normal samples, since our objective is to maximize the likelihood of the normal data distribution. We also provide sensitivity analysis of balancing the loss terms and error bound analysis in Appendix D.

**Anomaly Scoring.**  We define the anomaly score for sample $x$ as the complement of its log-likelihood, where higher values correspond to stronger deviations from the distribution of normal samples. Owing to the monotonicity of the exponential function $exp(\cdot)$, the anomaly score $s$ for a sample $x$ can be equivalently written as:

$$s(x) = 1 - \exp(\log[p(x)]) \tag{8}$$

## 5 Experiments

**Datasets.**  We conduct experiments on datasets with real-world labeled anomalies as well as on benign datasets with synthetic anomalies. The datasets with real-world labeled anomalies include Wikipedia, Reddit, and MOOC, while the benign datasets include Enron, UCI, and LastFM. Detailed descriptions, statistics, and preprocessing procedures are provided in Appendix A.

**Baselines.**  We compare our framework against a diverse set of baselines for DGAD, which fall into three categories. (1) DTDG-based methods designed for DGAD, including AddGraph (Zheng et al., 2019), StrGNN (Cai et al., 2021), TADDY (Liu et al., 2023) and Netwalk (Yu et al., 2018). (2) CTDG-based methods designed for DGAD, including SAD (Tian et al., 2023) and GeneralDyG (Yang et al., 2024) (3) Representative CTDG models originally proposed for tasks such as link prediction but readily adaptable to anomaly detection, including JODIE (Kumar et al., 2019), DyRep (Trivedi et al., 2019), TGAT (Xu et al., 2020), TGN (Rossi et al., 2020), TCL (Wang et al., 2021a), GraphMixer (Cong et al., 2023), CAWN (Wang et al., 2021b), DyGFormer (Yu et al., 2023), and FreeDyG (Tian et al., 2024b). We provide detailed descriptions in Appendix B.

**Experiment Settings.**  Previous studies rely exclusively on Area Under the Receiver Operating Characteristic Curve (AUROC) as the evaluation metric. While useful as a ranking metric, AUROC fails to assess a model's capacity to clearly classify individual samples. In practice, it can remain high remain high even when anomaly score of samples collapse into an indistinguishable low-valued range as shown in Fig 1. To provide a comprehensive evaluation, we report results on three metrics: AUROC, F1 score and Average Precision (AP). All models are trained for up to 200 epochs with early stopping (patience = 10) and the checkpoint achieving the best validation performance is selected for testing. The batch size is fixed at 200 for all methods and datasets. Each experiment is repeated five times and the average performance is reported to mitigate randomness. We perform grid search over some hyper-parameters. The learning rate is varied in $\{1e-3, 1e-4, 1e-5\}$, weight decay in $\{1e-1, 1e-2, 1e-3, 1e-4, 1e-5\}$.

### 5.1 Main Results

Table 1 reports the results across all baselines on the datasets with real-world labeled anomalies. From the results, we can observe that DTDG-based methods consistently yield the worst performance across all datasets. This is because their reliance on snapshot-level modeling inevitably discards fine-grained temporal information, which are essential for accurately capturing anomalous behaviors in dynamic graphs. By contrast, both CTDG-based DGAD methods and representative CTDG models achieve substantially stronger results. Notably, the performance gap between them

| Methods | Wikipedia | | | Reddit | | | MOOC | | |
|---|---|---|---|---|---|---|---|---|---|
| | AUROC | AP | F1 | AUROC | AP | F1 | AUROC | AP | F1 |
| Netwalk | 73.10±2.12 | 1.28±1.14 | 0±0 | 59.18±2.02 | 0.09±0.04 | 0±0 | 64.12±0.98 | 2.21±0.43 | 0±0 |
| AddGraph | 74.80±1.98 | 1.63±1.19 | 0.92±0.15 | 58.37±4.28 | 0.12±0.05 | 0±0 | 66.35±1.76 | 2.52±0.39 | 0±0 |
| STRGNN | 72.87±3.31 | 2.24±2.01 | 0±0 | 59.26±3.14 | 0.10±0.03 | 0±0 | 63.47±2.05 | 1.98±0.37 | 0±0 |
| Taddy | 75.40±2.88 | 2.52±1.41 | 1.34±0.17 | 61.04±2.33 | 0.14±0.06 | 0.10±0.05 | 67.02±1.64 | 3.83±0.41 | 0±0 |
| SAD | 79.84±1.91 | 5.12±2.03 | 4.25±3.84 | 62.98±2.05 | 0.17±0.05 | 0.76±0.48 | 71.25±0.77 | 6.74±0.52 | 3.12±1.21 |
| GeneralDyG | 77.52±1.05 | 3.15±0.87 | 1.06±1.41 | 61.43±1.48 | 0.15±0.03 | 0±0 | 70.12±0.83 | 5.31±0.64 | 0±0 |
| JODIE | 80.23±1.39 | 1.87±1.02 | 0±0 | 55.93±5.06 | 0.13±0.03 | 0±0 | 72.12±0.84 | 2.40±0.64 | 0±0 |
| DyRep | 83.89±1.03 | 2.60±0.31 | 0±0 | 58.83±2.95 | 0.14±0.02 | 0±0 | 72.21±0.25 | 3.56±0.04 | 0±0 |
| TGN | 85.51±1.12 | 3.80±1.47 | 0.96±1.32 | 64.31±0.72 | 0.14±0.01 | 0±0 | 76.63±0.98 | 6.47±0.15 | 0±0 |
| TGAT | 76.93±1.14 | 2.78±1.04 | 0.46±1.03 | 61.58±1.72 | 0.13±0.01 | 0±0 | 69.05±0.92 | 4.86±0.41 | 0±0 |
| TCL | 77.69±0.74 | 5.38±1.21 | 2.41±0.0 | 60.47±2.13 | 0.22±0.21 | 0±0 | 72.51±1.76 | 7.87±0.71 | 0±0 |
| CAWN | 78.97±0.56 | 4.96±0.72 | 1.44±1.32 | 65.29±0.92 | 0.18±0.03 | 0.14±0.32 | 72.63±0.39 | 7.58±0.24 | 0±0 |
| GraphMixer | 76.19±2.29 | 2.80±1.65 | 1.67±3.73 | 60.11±3.61 | 0.13±0.02 | 0±0 | 71.03±0.52 | 5.32±1.07 | 0±0 |
| DyGFormer | 85.58±1.25 | 2.58±0.72 | 0.48±1.06 | 66.70±2.09 | 0.25±0.11 | 0±0 | 72.63±0.33 | 6.20±0.36 | 0±0 |
| FreeDyG | 77.22±4.21 | 3.01±1.19 | 1.26±0.73 | 63.99±2.76 | 0.19±0.04 | 0±0 | 73.10±0.71 | 5.91±0.92 | 0±0 |
| **SDGAD (TCL)** | 80.36±0.69 | **5.41±1.23** | **8.34±2.72** | 62.22±0.95 | 0.49±0.26 | 2.74±1.86 | 72.87±1.40 | 7.89±0.38 | **7.15±0.64** |
| **SDGAD (CAWN)** | 80.84±0.65 | 5.15±0.79 | 7.41±3.04 | 66.81±1.08 | 0.57±0.05 | **3.28±2.02** | 73.02±0.41 | **8.62±0.25** | 6.74±0.53 |
| **SDGAD (DyGFormer)** | **86.60±1.20** | 3.71±0.77 | 4.15±2.06 | **67.24±1.12** | **0.88±0.11** | 3.15±1.11 | **73.25±0.36** | 6.39±0.48 | 5.86±0.77 |

Table 1: Performance comparison on datasets with real anomalies. Results are mean ± standard deviation, with all values scaled by 100. The best metric is highlighted in bold.

remains small, suggesting that task-specific designs for DGAD add only limited benefits. A key reason is that both categories concentrate on aggregating temporal structural information to obtain expressive representations, while giving less emphasis to constructing clear and robust decision boundaries for anomaly detection. By comparison, our framework consistently outperforms all competing baselines. For example, when built on top of TCL, our framework improves F1 by a large margin while maintaining strong AUROC and AP performance. Similar gains are observed when using other CTDG backbones such as CAWN and DyGFormer, indicating that the improvements are not tied to a specific encoder design. On average, SDGAD yields significant improvements over the best competing baselines across all metrics, confirming its effectiveness.

We also experiments on datasets with synthetic anomalies, with the result shown in Table 2. Note that synthetic anomalies may contain both normal and abnormal interactions for the same node pair within a single snapshot. Since DTDG methods discard temporal order, they cannot resolve such conflicts and are therefore unsuitable. The results again support our findings: general CTDG models often achieve reasonable AUROC but suffer from low F1, reflecting weak decision boundaries. In contrast, SDGAD consistently delivers substantial gains across all backbones, markedly boosting F1 while maintaining or improving AUROC and AP. The improvements hold for TCL, CAWN, and DyGFormer, with the latter achieving the strongest overall results. Taken together, these results demonstrate that SDGAD complements diverse CTDG models and consistently enhances their anomaly detection capability in a backbone-agnostic manner.

## 5.2 ABLATION STUDIES AND QUALITATIVE RESULTS

To verify the effectiveness of each design, we conduct ablation studies on the Wikipedia and MOOC datasets. We designed three variants including (1) **w/o Res** which removes the residual representation encoding and directly uses the representation computed with both historical and current interaction information. (2) **w/o** $\mathcal{L}_{\mathcal{RR}}$ which removes the restriction loss applied to the residual representations of normal samples. (3) **w/o** $\mathcal{L}_{\mathcal{BO}}$ which replaces the proposed bi-boundary optimization with single-boundary optimization.

As shown in Table 3, removing the residual representation **w/o Res** results in the most severe performance degradation: all metrics drop accompanied by a substantial increase in standard deviations. It demonstrates that residual representations play an indispensable role, as they encode discriminative signals that are essential for anomaly detection. For **w/o** $\mathcal{L}_{\mathcal{RR}}$, AUROC and AP remain close to those of the full model, while F1 declines markedly. The reason is that, without the restriction imposed

| Methods | UCI | | | Enron | | | LastFM | | |
|---|---|---|---|---|---|---|---|---|---|
| | AUROC | AP | F1 | AUROC | AP | F1 | AUROC | AP | F1 |
| SAD | 81.12±0.75 | 6.05±3.42 | 2.93±2.88 | 77.46±1.12 | 5.01±2.55 | 2.11±3.66 | 79.52±2.64 | 5.77±3.91 | 3.08±1.27 |
| GeneralDyG | 78.21±0.88 | 4.51±1.07 | 1.27±0.65 | 73.92±2.31 | 3.92±1.98 | 0.84±1.12 | 75.66±1.02 | 4.33±0.92 | 1.05±0.73 |
| JODIE | 75.08±3.11 | 3.06±0.74 | 0.62±0.48 | 70.14±1.42 | 2.77±0.81 | 0.38±0.33 | 72.41±2.08 | 3.12±0.66 | 0.71±0.57 |
| DyRep | 76.92±1.27 | 3.59±0.95 | 0.91±0.52 | 72.06±1.98 | 3.21±1.07 | 0.64±0.49 | 74.19±0.94 | 3.74±1.12 | 0.98±0.61 |
| TGN | 84.63±0.54 | 5.43±2.12 | 1.46±1.15 | 79.92±0.63 | 5.03±1.26 | 1.12±0.88 | 81.05±0.71 | 5.39±1.83 | 1.31±0.97 |
| TGAT | 79.51±1.72 | 4.65±1.08 | 1.08±0.91 | 75.77±1.06 | 4.06±0.92 | 0.95±0.74 | 78.21±0.93 | 4.98±1.35 | 1.24±0.88 |
| TCL | 81.56±0.91 | 7.18±2.98 | 2.24±2.63 | 77.41±1.42 | 5.84±2.01 | 1.73±1.54 | 79.61±1.15 | 5.97±1.74 | 2.01±1.92 |
| CAWN | 80.84±2.67 | 5.79±1.15 | 1.92±1.41 | 83.66±0.82 | 6.31±3.11 | 2.37±3.88 | 80.22±1.76 | 5.88±1.08 | 2.06±0.94 |
| GraphMixer | 79.74±1.03 | 4.86±3.01 | 1.57±2.04 | 76.23±1.61 | 4.39±0.97 | 1.31±0.76 | 78.96±2.55 | 5.32±3.42 | 1.69±2.08 |
| DyGFormer | 86.07±0.49 | 5.88±1.24 | 1.98±1.01 | 81.73±0.57 | 5.43±0.95 | 1.64±0.87 | 82.94±0.62 | 5.77±1.08 | 1.82±1.12 |
| FreeDyG | 81.29±1.45 | 5.03±2.63 | 1.71±1.99 | 77.12±0.96 | 4.72±1.04 | 1.39±0.85 | 78.86±1.58 | 4.97±1.73 | 1.61±1.42 |
| **SDGAD (TCL)** | 84.70±0.82 | **8.90±2.20** | **4.35±1.55** | 80.42±1.18 | 6.25±1.42 | 3.25±1.05 | 83.95±0.91 | **7.55±2.30** | **4.72±1.46** |
| **SDGAD (CAWN)** | 82.41±1.14 | 6.35±1.28 | 3.05±1.12 | 85.64±0.72 | **7.95±2.35** | **4.55±2.05** | 82.71±1.29 | 6.38±1.01 | 3.66±1.58 |
| **SDGAD (DyGFormer)** | **88.72±0.44** | 7.05±1.63 | 3.18±1.41 | **86.02±0.56** | 6.15±1.77 | 4.72±1.69 | **86.45±0.59** | 6.81±1.34 | 4.48±2.02 |

Table 2: Performance comparison on datasets with synthetic anomalies. Results are mean ± standard deviation, with all values scaled by 100. The best results are highlighted in bold.

on normal residuals, the residual space becomes scale-inconsistent, making it difficult to separate anomalies with small residuals. Consequently, although the relative ranking of samples is largely preserved (yielding stable AUROC and AP), the misaligned decision boundary causes a noticeable drop in F1. In the case of **w/o** $\mathcal{L}_{\mathcal{BO}}$, AUROC and AP change little, whereas F1 drops on average and fluctuates more. This suggests that although the overall score distribution is preserved, the lack of a decision margin makes boundary cases less stable. The bi-boundary optimization mitigates this issue by introducing such a margin, thereby reducing ambiguity.

To intuitively illustrate the effectiveness of each component, we visualize the log-likelihood distributions of normal and anomalous samples under different variants in Figure 3. The experiment is conducted on the *Wikipedia* dataset with TCL as the base encoder. In Figure 3(a), the left plot corresponds to baseline without any components, where the distributions of normal and anomalous samples heavily overlap, reflecting poor discriminability. When residual representations are included but the restriction is removed (**w/o** $\mathcal{L}_{\mathcal{RR}}$, Figure 3(a), right), the overlap is reduced, yet normal samples remain scattered. Figure 3(b), left, shows the effect of **w/o** $\mathcal{L}_{\mathcal{BO}}$. Here the normal distribution becomes more compact and separation improves, but some anomalies still cluster near the boundary due to the lack of explicit optimization in likelihood space. Finally, in Figure 3(b) right, the full framework yields a concentrated normal distribution at high-likelihood values, with anomalies clearly shifted toward low-likelihood regions, forming a sharp decision boundary.

| Variant | Wikipedia | | | MOOC | | |
|---|---|---|---|---|---|---|
| | AUROC | AP | F1 | AUROC | AP | F1 |
| SDGAD(TCL) | **80.36±0.69** | **5.41±1.23** | **8.34±2.72** | **72.87±1.40** | **7.89±0.38** | **7.15±0.64** |
| w/o Res | 77.79±3.17 | 3.83±2.48 | 5.29±5.69 | 69.14±2.48 | 5.14±2.48 | 6.58±2.48 |
| w/o $\mathcal{L}_{\mathcal{RR}}$ | 79.44±0.80 | 5.33±1.47 | 6.01±0.72 | 71.02±1.28 | 7.41±0.62 | 5.96±0.81 |
| w/o $\mathcal{L}_{\mathcal{BO}}$ | 80.36±1.00 | 5.38±1.32 | 7.41±3.29 | 72.54±1.35 | 7.56±0.55 | 6.42±0.97 |
| TCL | 77.69±0.74 | 5.28±1.21 | 2.41±0.00 | 72.51±1.76 | 7.87±0.71 | 0±0 |

Table 3: Ablation studies on Wikipedia and MOOC.

## 5.3 HYPER-PARAMETER STUDY

In this section, we analyze two hyperparameters that are critical to the performance of our framework. The first is $L$, which controls the number of sampled historical neighbors during the residual representation encoding stage. The second parameter is $\alpha$, which determines the position of the normal boundary as the $\alpha$-th percentile of the sorted normal log-likelihood distribution $\mathcal{D}_n$. We conduct experiments with SDGAD(TCL), varying $L \in 2, 5, 10, 20, 32$ and $\alpha \in 0.001, 0.005, 0.01, 0.05, 0.1$. The experimental results are summarized in Tables 4a and 4b.

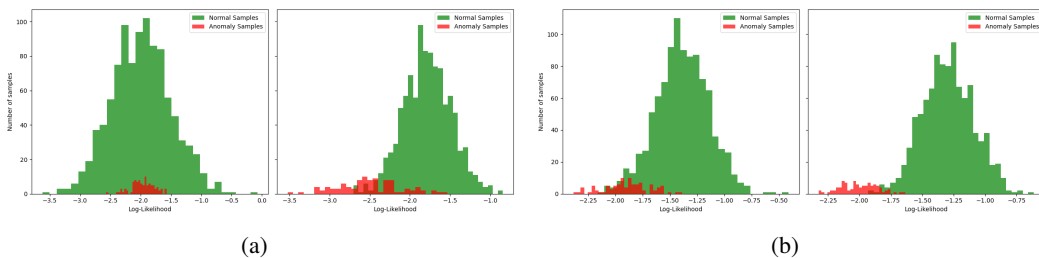

(a)                    (b)

Figure 3: Visualization of log-likelihood distributions under different ablation variants. (a) Left: baseline without any components; Right: adding residual representations but without restriction. (b) Left: with representation restriction but without bi-boundary optimization; Right: full framework.

| L | Wikipedia | | | MOOC | | | $\alpha$ | Wikipedia | | | MOOC | | |
|---|---|---|---|---|---|---|---|---|---|---|---|---|---|
| | AUROC | AP | F1 | AUROC | AP | F1 | | AUROC | AP | F1 | AUROC | AP | F1 |
| 2 | 80.36±0.69 | **5.41±1.23** | **8.34±2.72** | 65.75±0.72 | 3.26±0.19 | **8.74±0.54** | 0.001 | **81.12±1.24** | **5.68±1.61** | 5.40±3.10 | 71.28±1.42 | 6.93±0.95 | **7.42±2.81** |
| 5 | 83.76±0.49 | 5.36±0.53 | 5.33±3.23 | 68.51±1.86 | 3.33±0.16 | 4.27±1.83 | 0.005 | 80.77±0.84 | 5.68±1.09 | 4.68±1.83 | 71.88±0.96 | 7.40±0.27 | 6.65±0.66 |
| 10 | 85.94±0.74 | 4.72±1.33 | 4.43±3.44 | 70.31±0.29 | 6.46±0.56 | 7.46±0.79 | 0.01 | 80.36±0.69 | 5.41±1.23 | **8.34±2.72** | 72.87±1.40 | **7.89±0.38** | 7.15±0.64 |
| 20 | 87.22±0.76 | 4.94±0.85 | 4.67±4.04 | 72.87±1.40 | **7.89±0.38** | 7.15±0.64 | 0.05 | 79.87±0.42 | 5.39±1.09 | 6.37±3.91 | 72.91±0.45 | 7.12±0.40 | 5.43±0.43 |
| 32 | **88.06±0.38** | 5.04±1.55 | 2.92±1.59 | **73.56±1.11** | 7.24±0.44 | 6.20±0.38 | 0.1 | 79.70±0.89 | 5.37±0.91 | 6.46±2.68 | 72.73±0.81 | 7.00±1.03 | 5.78±1.04 |

(a) Performance with different values of $L$          (b) Performance with different values of $\alpha$

Table 4: Effect of hyperparameters $L$ and $\alpha$ on performance. The best results are highlighted in bold and the underlined results correspond to those reported in the main results Table. 1.

For hyperparameter $L$, we can observe a consistent trend: as $L$ increases, AUROC gradually improves while F1 consistently decreases. This phenomenon aligns well with the characteristics of residual representation. When more historical information is aggregated during representation encoding, the residual signals become diluted, thereby reducing the discriminative capacity between normal and abnormal samples and directly lowering F1 performance. At the same time, AUROC remains less sensitive to this dilution because it evaluates ranking consistency rather than absolute separability. Even when anomaly and normal scores converge and exhibit weaker discriminability, as long as anomalies tend to be ranked above normal samples, AUROC will continue to increase. This explains why larger $L$ yields higher AUROC but lower F1.

The results for hyperparameter $\alpha$ highlight the importance of selecting an appropriate boundary. A large $\alpha$ shifts the normal boundary closer to the center of the normal distribution, improving generalization but weakening discrimination and misclassifying borderline normal samples. In contrast, a small $\alpha$ enforces an overly strict boundary, increasing the risk of overfitting. Empirically, on the Wikipedia dataset, AUROC achieves its maximum at $\alpha$=0.001, while F1 peaks at $\alpha$=0.01. On the MOOC dataset, AUROC is highest at $\alpha$=0.01, whereas F1 is maximized at $\alpha$=0.001. Overall, the results demonstrate a trade-off, where small $\alpha$ favors discriminability and large $\alpha$ favors robustness, with $\alpha = 0.01$ offering the best balance. We further provide a more comprehensive hyper-parameter analysis in the Appendix C.

## 6 CONCLUSION

In this paper, we introduced SDGAD, a supervised framework for dynamic graph anomaly detection that is both effective and generalizable. Our framework learns informative and discriminative representations through a residual representation with a restriction mechanism. Furthermore, we employ a normalizing flow to model the log-likelihood distribution of normal samples, enabling the detection of anomalies as out-of-distribution instances. Building on this distribution, we derive an explicit decision boundary and introduce a bi-boundary optimization strategy to further enhance generalization. Extensive evaluations demonstrate the superiority of our framework.

## 7 ETHICS STATEMENT

This work adheres to the ICLR Code of Ethics. In this study, no human subjects or animal experimentation was involved. All datasets used were sourced in compliance with relevant usage guidelines, ensuring no violation of privacy. We have taken care to avoid any biases or discriminatory outcomes in our research process. No personally identifiable information was used, and no experiments were conducted that could raise privacy or security concerns. We are committed to maintaining transparency and integrity throughout the research process.

## 8 REPRODUCIBILITY STATEMENT

We have made every effort to ensure that the results presented in this paper are reproducible. All code and datasets have been made publicly available in an anonymous repository to facilitate replication and verification. The experimental setup, including training steps, model configurations, and hardware details, is described in detail in the paper. We believe these measures will enable other researchers to reproduce our work and further advance the field.

## 9 LLM USAGE

Large Language Models (LLMs) were used to aid in the writing and polishing of the manuscript. Specifically, we used an LLM to assist in refining the language, improving readability, and ensuring clarity in various sections of the paper. The model helped with tasks such as sentence rephrasing, grammar checking, and enhancing the overall flow of the text.

It is important to note that the LLM was not involved in the ideation, research methodology, or experimental design. All research concepts, ideas, and analyses were developed and conducted by the authors. The contributions of the LLM were solely focused on improving the linguistic quality of the paper, with no involvement in the scientific content or data analysis.

The authors take full responsibility for the content of the manuscript, including any text generated or polished by the LLM. We have ensured that the LLM-generated text adheres to ethical guidelines and does not contribute to plagiarism or scientific misconduct.

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

# A DETAILS OF DATASETS

| Dataset | Nodes | Edges | Unique Edges | Timesteps | Edge Feature | Anomaly Ratio | Density |
|---|---|---|---|---|---|---|---|
| Wikipedia | 9227 | 157474 | 18257 | 152757 | 172 | 0.14% | 4.30E-03 |
| Reddit | 10984 | 672447 | 78516 | 669065 | 172 | 0.05% | 8.51E-03 |
| MOOC | 7144 | 411749 | 178443 | 345600 | 4 | 0.99% | 1.26E-02 |
| LastFm | 1980 | 1293103 | 154993 | 1283614 | 0 | - | 5.57E-01 |
| Enron | 184 | 125235 | 3125 | 22632 | 0 | - | 5.53E+00 |
| UCI | 1899 | 59835 | 20296 | 58911 | 0 | - | 3.66E-02 |

Table 5: Dataset statistics

## A.1 DESCRIPTION OF DATASETS

**Wikipedia**: A bipartite graph of user edits on Wikipedia pages, where nodes represent users and pages, and edges denote timestamped edits. Each interaction is associated with a 172-dimensional LIWC feature. Dynamic labels indicate whether the corresponding edit behavior is banned.

**Reddit**: A bipartite dataset of user posts on Reddit over one month. Nodes correspond to users and subreddits, with timestamped posting edges and 172-dimensional LIWC features. Dynamic labels indicate whether the corresponding post behavior is banned.

**MOOC**: A bipartite interaction network between students and course units (e.g., videos, problem sets). Edges represent student access behaviors with 4-dimensional features. Dynamic labels indicate whether the access behavior is banned.

**LastFM**: A bipartite graph of music listening activities over one month, where nodes are users and songs, and edges denote timestamped listening events.

**Enron**: An email communication network with about 50K messages exchanged among Enron employees over three years. No attributes are provided.

**UCI**: A communication network among students at the University of California, Irvine, with timestamped interactions at second-level granularity. No additional features are included.

## A.2 PREPROCESSING PROCEDURES

We split all datasets into three chronological segments for training, validation, and testing with ratios of 40%-20%-40%. For the three datasets with real anomalies, the anomaly proportion remains relatively consistent across the training, validation, and testing sets. Specifically, For the Wikipedia dataset, the anomaly ratio remains very stable across subsets, with anomaly ratio of approximately $0.14\%$ in train, $0.15\%$ in validation, and $0.13\%$ in test. For the MOOC dataset, the anomaly proportion is slightly higher overall, with anomaly ratio of about $1.14\%$ in train, $0.86\%$ in validation, and $0.9\%$ in test. For the Reddit dataset, anomalies are extremely sparse, with anomaly ratio of $0.024\%$ in train, $0.065\%$ in validation, and $0.08\%$ in test.

For the three datasets (LastFM, Enron, UCI) that do not contain real anomalies, we follow the anomaly synthesis strategy for dynamic graphs proposed by Postuvan et al. (2024) to inject synthetic anomalies. Specifically, Postuvan et al. (2024) introduce five strategies based on three fundamental anomaly types: (i) randomizing the destination node to create structural anomalies, (ii) randomizing attributes to create contextual anomalies, and (iii) randomizing edge timestamps to create temporal anomalies. Since these three datasets do not contain original node/edge attributes, the second type is not applicable. We therefore adopt only (i) and (iii) and construct three synthetic anomaly types: T (temporal anomaly), S (structural anomaly), and T-S (temporal and structural anomaly). To evaluate the generalization ability of our model, we inject 1% T anomalies into the training and validation sets, while in the test set we inject 0.33% of each anomaly type (T, S, T-S).

# B BASELINES

## B.1 DESCRIPTION OF BASELINES

**Netwalk** (Yu et al., 2018)    A temporal random-walk embedding model that learns joint node–edge representations. It maintains an online clustering of embedding trajectories and flags deviations as anomalies.

**AddGraph** (Zheng et al., 2019)    A semi-supervised DTDG method that augments a temporal GCN with attention to capture long- and short-term patterns, and trains with selective negative sampling plus a margin loss to address label sparsity.

**StrGNN** (Cai et al., 2021)    A subgraph-based temporal model for edge anomalies. It extracts the $h$-hop enclosing subgraph, labels node roles, applies graph convolution with SortPooling to obtain fixed-size snapshot features, and uses a GRU to capture temporal dynamics.

**Taddy** (Liu et al., 2023)    A dynamic-graph transformer with a learnable node encoding that separates global spatial, local spatial, and temporal terms. It samples edge-centered temporal substructures and uses attention to couple structural and temporal dependencies end to end.

**SAD** (Tian et al., 2023)    A semi-supervised CTDG method for DGAD. It first predicts node-level anomaly scores and stores score–timestamp pairs in a memory bank to estimate a normal prior and apply a deviation loss, and adds a pseudo-label contrastive module that forms score-based pseudo-groups and treats intra-group pairs as positives.

**GeneralDyG** (Yang et al., 2024)    An unsupervised CTDG method for DGAD. It uses a GNN extractor that embeds nodes, edges, and topology and alternates node- and edge-centric message passing. It inserts special tokens into feature sequences to encode hierarchical relations between anomalous events while balancing global temporal context and local dynamics, and trains on ego-graph samples of anomalous events to reduce computation.

**JODIE** (Kumar et al., 2019)    An RNN-based model. For each interaction between $v_i$ and $v_j$ at time $t$, it updates the temporal embedding of $v_i$ using its previous state, the latest state of $v_j$, the link features, and the time gap since the last interaction. Final embeddings are extrapolated via a linear projection on the last observed state.

**DyRep** (Trivedi et al., 2019)    An RNN-based model with attention and a temporal point-process head. After each event, node states are recurrently updated while attention aggregates neighbor context, and the conditional intensity models event timing.

**TGAT** (Xu et al., 2020)    A temporal graph attention model that incorporates both structural and temporal signals. Each node feature is concatenated with a trainable time encoding. Multi-head self-attention is then applied over temporal neighbors to compute node representations.

**TGN** (Rossi et al., 2020)    A memory-augmented temporal GNN model which integrates a memory mechanism with self-attention. Each node maintains a memory state that summarizes its history. Upon observing an interaction, the states of involved nodes are updated through an RNN-based memory updater. Final embeddings are computed by aggregating K-hop temporal neighborhoods with self-attention.

**GraphMixer** (Cong et al., 2023)    A lightweight MLP-Mixer architecture. It adopts a fixed (non-trainable) time encoding function, integrates it into an MLP-Mixer for edge encoding, and summarizes neighbor information through mean pooling.

**TCL** (Wang et al., 2021a)    A transformer-based temporal graph model. It builds node interaction sequences via breadth-first traversal on temporal subgraphs, applies a graph transformer to capture joint structural–temporal dependencies, and uses cross-attention to model inter-node interactions.

**CAWN** (Wang et al., 2021b)    Combines RNNs and self-attention via temporal random walks. It replaces raw node identities with hitting counts obtained from sampled walks, encodes these motifs with RNNs, and aggregates multiple walks into a single node representation using self-attention.

**DyGFormer** (Yu et al., 2023)    A transformer operating on patched interaction sequences. It segments each node's timeline into patches and learns temporal dependencies across patches to obtain node representations.

**FreeDyG** (Tian et al., 2024b)   A frequency-aware temporal graph model. It first encodes the time, node and edge information, augmented with a node-pair frequency encoding mechanism. A frequency-enhanced MLP-Mixer is then applied to capture periodicities and temporal shifts, then inverts and mixes to yield frequency-salient embeddings.

### B.2   IMPLEMENTATION DETAILS

We employ several CTDG-based methods that were originally proposed for fundamental dynamic graph tasks such as link prediction and node classification. Owing to the close connection between these tasks and anomaly detection, such methods can be extended to the anomaly detection setting. The only required modification concerns the treatment of event samples: in link prediction, the target edge of the current interaction cannot be used as input, whereas in anomaly detection it can. Thus, we follow Postuvan et al. (2024) and apply this adjustment to adapt general CTDG-based methods for anomaly detection.

## C   HYPER-PARAMETERS STUDIES

We further study two important hyperparameters to analyze their effect on performance of SDGAD. The first is the decision margin $\tau$, which is used in bi-boundary optimization. After the normal boundary is determined by $\alpha$, the anomaly boundary is defined as $B_n - \tau$. Since the log-likelihood values are rescaled into the range $[-1, 0]$ by a normalization constant for stable optimization, $\tau$ can also be treated as a constant in $[0, 1]$. The second hyperparameter is the interval $[r_{\max}, r_{\min}]$ used in the representation restriction phase. According to Eq. 5, $A$ is normalized to a constant range, $r_{\max}$ and $r_{\min}$ can also be treated as constants. Importantly, what matters is the interval between them rather than their absolute values. In our experiments, we fix $r_{\max} = 0.4$ and determine $r_{\min}$ through a coefficient parameter, i.e., $r_{\min} = \text{coefficient} \times r_{\max}$. Therefore, we focus on exploring the effect of the coefficient (coe) on model performance. Specifically, we conduct experiments with SDGAD(TCL), varying $\tau \in 0.05, 0.1, 0.15, 0.2$ and $coe \in 0.99, 0.95, 0.90, 0.80$. The experimental results are summarized in Tables 6 and 7.

The results in Table 6 show that the decision margin $\tau$ has a non-trivial impact on performance. A proper margin provides sufficient separation between normal and anomalous boundaries, while margins that are too narrow or too wide both lead to performance degradation. This reflects the central role of $\tau$ in balancing discrimination against robustness during boundary optimization. Specifically, $\tau = 0.1$ consistently delivers the strongest results across all metrics. A too-narrow margin (e.g., $\tau = 0.05$) compresses the space between normal and anomaly boundaries, leaving insufficient room for effective separation and thereby depressing F1. In contrast, an overly wide margin (e.g., $\tau = 0.2$) relaxes the anomaly boundary excessively, which introduces noise and wrongly pushes borderline normal samples into the anomaly region. Interestingly, $\tau = 0.15$ offers moderate AUROC but still struggles on F1, showing that ranking consistency can be preserved even when classification precision is compromised. For the $coe$ in Table 7, we observe a similar trade-off. A tighter interval enforces stronger consistency among normal samples and thus favors discriminability, whereas a looser interval introduces flexibility but weakens the separation from anomalies. The results suggest that carefully tuning this interval is crucial for achieving a good balance between precision and generalization.

| $\tau$ | Wikipedia | | | MOOC | | |
|---|---|---|---|---|---|---|
| | AUROC | AP | F1 | AUROC | AP | F1 |
| 0.05 | 80.26±0.95 | 4.52±1.21 | 5.06±3.49 | 71.92±1.10 | 7.10±0.18 | 5.77±0.85 |
| 0.1 | **80.36±0.69** | **5.41±1.23** | **8.34±2.72** | **72.87±1.40** | **7.89±0.38** | **7.15±0.64** |
| 0.15 | 80.18±1.15 | 5.08±1.60 | 5.72±2.62 | 72.32±1.22 | 6.67±1.13 | 5.77±1.44 |
| 0.2 | 77.26±6.33 | 3.57±2.17 | 5.07±3.57 | 72.02±1.53 | 7.01±0.37 | 5.91±1.22 |

Table 6: Effect of hyperparameter $\tau$ on performance. The best results are highlighted in bold and the underlined results correspond to those reported in the Table. 1.

| $coe$ | Wikipedia | | | MOOC | | |
|---|---|---|---|---|---|---|
| | AUROC | AP | F1 | AUROC | AP | F1 |
| 0.99 | 80.36±0.69 | 5.41±1.23 | **8.34±2.72** | 72.87±1.40 | **7.89±0.38** | **7.15±0.64** |
| 0.95 | 80.29±1.74 | 5.28± 1.76 | 3.51±0.81 | 72.57±0.48 | 7.18±0.34 | 7.02±0.80 |
| 0.90 | 80.77±0.72 | **5.57±1.69** | 5.79±1.85 | **72.98±0.95** | 7.24±0.44 | 6.02±0.29 |
| 0.80 | **81.15±0.68** | 5.43±1.70 | 3.97±0.67 | 72.63±0.87 | 7.40±0.27 | 6.65±0.66 |

Table 7: Effect of hyperparameter *coe* on performance. The best results are highlighted in bold and the underlined results correspond to those reported in the Table. 1.

# D  LOSS ANALYSIS

## D.1  SENSITIVITY OF BALANCING THE LOSS TERMS

Our framework jointly optimizes three objectives: the maximum likelihood loss $\mathcal{L}_{\mathcal{ML}}$, the restriction loss $\mathcal{L}_{RR}$, and the bi-boundary optimization loss $\mathcal{L}_{\mathcal{BO}}$. Since $\mathcal{L}_{\mathcal{ML}}$ is the basic training objective for the normalizing flow, we fix its weight to 1 and introduce $\lambda_1$ and $\lambda_2$ as the weights for $\mathcal{L}_{\mathcal{BO}}$ and $\mathcal{L}_{RR}$, respectively. Table 8 reports the results of varying these coefficients. Overall, performance remains relatively stable when $\lambda_1$ and $\lambda_2$ are set within a moderate range (e.g., $1/0.1$, $1/0.5$, $1/0.7$). On Wikipedia, $\mathcal{L}_{\mathcal{BO}}$ plays a central role, as reducing its weight ($\lambda_1 = 0.5$) causes substantial performance drops, whereas settings with $\lambda_1 = 1$ achieve consistently higher AUROC and F1. By contrast, on MOOC, $\lambda_1 = 0.5$ yields the strongest results, suggesting that a lighter weight on $\mathcal{L}_{\mathcal{BO}}$ helps avoid overfitting. For $\mathcal{L}_{\mathcal{RR}}$, moderate changes in $\lambda_2$ generally have a weaker effect, but excessive weighting ($\lambda_2 = 2$) degrades performance across both datasets, likely due to over-constraining the residual space.

Notably, there is no configuration simultaneously maximizes all metrics. For example, $\lambda_1 = 1, \lambda_2 = 0.1$ achieves the best AUROC on Wikipedia, while $\lambda_1 = 1, \lambda_2 = 0.5$ yields higher AP and F1 with more stable variance. Thus, we adopt $\lambda_1 = 1, \lambda_2 = 0.5$ as the best setting for wikipedia. This choice reflects a deliberate trade-off: the reported results in our experiments are not necessarily the absolute optimum for any single metric, but rather represent a balanced configuration that ensures stable and robust performance across all evaluation metrics.

| $\lambda_1/\lambda_2$ | Wikipedia | | | MOOC | | |
|---|---|---|---|---|---|---|
| | AUROC | AP | F1 | AUROC | AP | F1 |
| 0.5/0.5 | 71.73±8.80 | 4.08±3.17 | 4.18±3.93 | **72.87±1.40** | **7.89±0.38** | 7.15±0.64 |
| 1/0.1 | **80.67±1.14** | 4.47±0.74 | 8.32±2.00 | 70.02±1.15 | 5.22±1.26 | 4.74±4.92 |
| 1/0.5 | 80.36±0.69 | 5.41±1.23 | 8.34±2.72 | 70.48±0.76 | 5.37±1.14 | **7.69±4.52** |
| 1/0.7 | 80.64±0.47 | 5.33±1.47 | 6.41±3.08 | 66.10±7.71 | 4.30±2.74 | 4.90±5.27 |
| 1/1 | 76.65±9.03 | 4.71±2.19 | 6.78±4.27 | 70.87±0.51 | 5.45±0.73 | 6.24±2.01 |
| 1/2 | 74.91±8.65 | 4.01±2.90 | 4.65±4.48 | 70.75±0.82 | 5.80±1.54 | 6.62±1.18 |

Table 8: Results when varying different $\lambda_1/\lambda_2$ values for balancing loss terms.

## D.2  ERROR BOUND ANALYSIS

**Proposition 1.** *Assume that* $\Phi_{\theta^*} \in \arg\min_{\theta \in \Theta}\{\mathcal{L}_{\mathcal{ML}} + \lambda_1 \mathcal{L}_{\mathcal{BO}}\}$. *That is,* $\Phi_{\theta^*}$ *corresponds to the optimal parameters minimizing the joint objective of the maximum-likelihood loss and the bi-boundary optimization loss. Then we have that*

$$\mathbb{E}_{y_i=0}[\max((\mathcal{B}'_n - \log[p(x_i)]), 0)] + \mathbb{E}_{y_j=1}[\max((\log[p(x_j)] - \mathcal{B}'_a), 0)]$$

$$\leq (\mathcal{B}_n - \mathcal{B}_a)\mathcal{L}_{\mathcal{BO}}(\Phi_{\theta^*}) + \frac{N}{(N+M)}[\max(1 + \mathcal{B}'_n, -\mathcal{B}'_a)]$$

$$\leq \frac{(\frac{d}{2}\log(2\pi) - \frac{1}{2})(\mathcal{B}_n - \mathcal{B}_a)}{\lambda} + \frac{N}{(N+M)} \tag{9}$$

*where* $y = 0$, $y = 1$ *denote normal and anomalous labels,* $\mathcal{B}'_n = \mathcal{B}_n - \epsilon, \mathcal{B}'_a = \mathcal{B}_a + \epsilon, \epsilon \in (0, \mathcal{B}_n - \mathcal{B}_a)$, $N$ *and* $M$ *are the number of normal and abnormal samples.*

*proof.* Suppose we sort all samples (both normal and anomalous) by their log-likelihoods in descending order: $\log[p(x_1)] \geq \log[p(x_2)] \geq \cdots \geq \log[p(x_{N+M})]$. Let $\mathcal{B}_n = \log[p(x_N)]$ denote the normal boundary induced by the $N$-th ranked sample, which corresponds to the threshold for classifying normal sample. Under a worst-case assumption, all top-$N$ samples (which ideally should be normal) are misclassified, while the remaining $M$ anomalous samples have log-likelihoods lying between $\mathcal{B}_a$ and $\mathcal{B}_n$. In this scenario, the expected margin-violation error can be bounded as:

$$\mathbb{E}_{y_i=0}[\max((\mathcal{B}'_n - \log[p(x_i)]), 0)] + \mathbb{E}_{y_j=1}[\max((\log[p(x_j)] - \mathcal{B}'_a), 0)]$$

$$\leq (\mathcal{B}'_n - \mathcal{B}'_a)\mathcal{L}'_{\mathcal{BO}}(\mathbf{\Phi}_{\theta^*}) + \frac{N}{(N+M)}[\max(1 + \mathcal{B}'_n, -\mathcal{B}'_a)]$$

$$\leq (\mathcal{B}_n - \mathcal{B}_a)\mathcal{L}_{\mathcal{BO}}(\mathbf{\Phi}_{\theta^*}) + \frac{N}{(N+M)} \tag{10}$$

Here $\mathcal{L}'_{\mathcal{BO}}$ denotes the $\ell_0$ norm based formulation of $\mathcal{L}_{\mathcal{BO}}$, which counts the number of samples violating the boundary constraints (i.e., the number of misclassified samples). It represents an idealized, non-differentiable version of $\mathcal{L}_{\mathcal{BO}}$, used only for theoretical analysis. The second inequality is obtained as $1 + \mathcal{B}'_n \leq 1$ and $-\mathcal{B}'_a \leq 1$ when $-1 \leq \mathcal{B}'_a < \mathcal{B}'_n \leq 0$ satisfies. Since $\mathbf{\Phi}_{\theta^*}$ is defined as the optimal parameter of the joint objective $\mathcal{L}_{\mathcal{ML}} + \lambda_1 \mathcal{L}_{\mathcal{BO}}$, its objective value cannot be larger than that of any other candidate solution. In particular, consider an arbitrary reference solution $\mathbf{\Phi}_{\theta'}$ such that $\mathcal{L}_{\mathcal{BO}}(\mathbf{\Phi}_{\theta'}) = 0$. By the optimality of $\mathbf{\Phi}_{\theta^*}$, we have:

$$\mathcal{L}_{\mathcal{ML}}(\mathbf{\Phi}_{\theta^*}) + \lambda_1 \mathcal{L}_{\mathcal{BO}}(\mathbf{\Phi}_{\theta^*}) \leq \mathcal{L}_{\mathcal{ML}}(\mathbf{\Phi}_{\theta'}) + \lambda_1 \mathcal{L}_{\mathcal{BO}}(\mathbf{\Phi}_{\theta'})$$
$$= \mathcal{L}_{\mathcal{ML}}(\mathbf{\Phi}_{\theta'}) \tag{11}$$

We isolate $\mathcal{L}_{\mathcal{BO}}(\mathbf{\Phi}_{\theta^*})$ as:

$$\mathcal{L}_{\mathcal{BO}}(\mathbf{\Phi}_{\theta^*}) \leq \frac{(\mathcal{L}_{\mathcal{ML}}(\mathbf{\Phi}_{\theta'}) - \mathcal{L}_{\mathcal{ML}}(\mathbf{\Phi}_{\theta^*}))}{\lambda_1}$$

$$\leq \frac{\left(\frac{1}{2}\mathbf{\Phi}_{\theta'}(x)^T\mathbf{\Phi}_{\theta'}(x) - \frac{1}{2}\mathbf{\Phi}_{\theta^*}(x)^T\mathbf{\Phi}_{\theta^*}(x) + \frac{1}{2}\mathbf{\Phi}_{\theta^*}(x)^T\mathbf{\Phi}_{\theta^*}(x) + \frac{d}{2}\log(2\pi)\right)}{\lambda_1}$$

$$\leq \frac{\frac{d}{2}\log(2\pi) - \frac{1}{2}}{\lambda_1} \tag{12}$$

To obtain a tractable bound, we assume a worst-case initialization:

$$\mathbf{\Phi}_{\theta'}(x)^T\mathbf{\Phi}_{\theta'}(x) = -1 \tag{13}$$

This assumption gives the largest possible gap between $\mathbf{\Phi}_{\theta'}$ and $\mathbf{\Phi}_{\theta^*}$, and thus produces the loosest valid bound. By combining the above E.q.(10) and E.q.(12), we have that

$$\mathbb{E}_{y_i=0}[\max((\mathcal{B}'_n - \log[p(x_i)]), 0)] + \mathbb{E}_{y_j=1}[\max((\log[p(x_j)] - \mathcal{B}'_a), 0)]$$

$$\leq \frac{(\frac{d}{2}\log(2\pi) - \frac{1}{2})(\mathcal{B}_n - \mathcal{B}_a)}{\lambda_1} + \frac{N}{(N+M)} \tag{14}$$

The above proposition highlights both the necessity and the effectiveness of the bi-boundary optimization loss $\mathcal{L}_{\mathcal{BO}}$. Ideally, increasing the weight $\lambda_1$ of $\mathcal{L}_{\mathcal{BO}}$ facilitates the convergence of the error bound toward zero. Moreover, the proposition implies that the presence of more anomalous samples can further enhance the reliability of discriminating between normal and abnormal samples.

## D.3 ERROR BOUND ANALYSIS UNDER THE FULL OBJECTIVE

We now make explicit how the representation restriction loss $L_{RR}$ contributes to the error through the geometry of the projected residuals. Recall that the restriction loss is

$$L_{RR} = (1 - y)\,\ell_{\text{int}}(x) + y\,\ell_{\text{anom}}(x)$$

where $y \in \{0, 1\}$ is the label, and for normal samples ($y = 0$) the *interval penalty* is

$$\ell_{\text{int}}(x) = \max(0, A(x) - r_{\max}) + \max(0, r_{\min} - A(x))$$

The anomalous part $\ell_{\mathrm{anom}}(x)$ is always non-negative. We denote the expected interval penalty on normal samples by

$$L_{RR}^{(0)} = \mathbb{E}_{y=0}\big[\ell_{\mathrm{int}}(x)\big].$$

By construction $L_{RR}^{(0)} \leq L_{RR}$, since $L_{RR}$ additionally includes the non-negative anomalous part. As in Appendix D.2, we define the margin–violation error as

$$\mathcal{E}_{\mathrm{mv}} = \mathbb{E}_{y_i=0}\big[\max(B_n' - \log[p(x_i)], 0)\big] + \mathbb{E}_{y_j=1}\big[\max(\log[p(x_j)] - B_a', 0)\big]$$

For notational convenience, we set $Z_i = \max(B_n' - \log[p(x_i)], 0)$ and $W_j = \max(\log[p(x_j)] - B_a', 0)$ so that $\mathcal{E}_{\mathrm{mv}} = \mathbb{E}_{y_i=0}[Z_i] + \mathbb{E}_{y_j=1}[W_j]$. Finally, let $C_B = \max(1 + B_n', -B_a')$ denote the constant used in Eq. (9) and we assume that the per-sample margin–violation on normal samples is uniformly bounded by a constant $C_{\mathrm{mv}} > 0$, that is,

$$0 \leq Z_i \leq C_{\mathrm{mv}} \quad \text{for all } i \text{ with } y_i = 0.$$

Then we can get the new proposition as follow:

**Proposition 2.** *Assume the setting of Proposition 1 and the uniform bound $0 \leq Z_i \leq C_{\mathrm{mv}}$ for all normal samples. Then for any slack parameter $\delta > 0$, the margin–violation error satisfies*

$$\mathcal{E}_{\mathrm{mv}} \leq (B_n - B_a) L_{\mathrm{BO}}(\Phi_{\theta^*}) + \frac{N}{N+M} C_B + \frac{C_{\mathrm{mv}}}{\delta} L_{RR} \tag{15}$$

*proof.* For a fixed $\delta > 0$, we define the event

$$\mathcal{G}_\delta(x) = \big\{ r_{\min} - \delta \leq A(x) \leq r_{\max} + \delta \big\}.$$

If $\mathcal{G}_\delta(x)$ does not hold for a normal sample, then either $A(x) > r_{\max} + \delta$ or $A(x) < r_{\min} - \delta$, and in both cases the distance from $A(x)$ to the interval $[r_{\min}, r_{\max}]$ is at least $\delta$. By the definition of $\ell_{\mathrm{int}}(x)$, this implies

$$\ell_{\mathrm{int}}(x) = \max(0, A(x) - r_{\max}) + \max(0, r_{\min} - A(x)) \geq \delta.$$

Equivalently,

$$\ell_{\mathrm{int}}(x) \geq \delta \cdot \mathbf{1}\{\neg\mathcal{G}_\delta(x)\} \quad \text{for } y = 0.$$

Taking expectations over normal samples gives

$$L_{RR}^{(0)} = \mathbb{E}_{y=0}\big[\ell_{\mathrm{int}}(x)\big] \geq \delta\, \mathbb{P}_{y=0}\big(\neg\mathcal{G}_\delta(x)\big),$$

and hence

$$\mathbb{P}_{y=0}\big(\neg\mathcal{G}_\delta(x)\big) \leq \frac{1}{\delta} L_{RR}^{(0)} \leq \frac{1}{\delta} L_{RR}. \tag{16}$$

We split the normal contribution according to the radius event $\mathcal{G}_\delta(x)$:

$$\mathbb{E}_{y_i=0}[Z_i] = \mathbb{E}_{y_i=0}\big[Z_i \mathbf{1}\{\mathcal{G}_\delta(x_i)\}\big] + \mathbb{E}_{y_i=0}\big[Z_i \mathbf{1}\{\neg\mathcal{G}_\delta(x_i)\}\big].$$

Using the uniform bound $0 \leq Z_i \leq C_{\mathrm{mv}}$ and Eq. 16, we obtain

$$\mathbb{E}_{y_i=0}[Z_i] \leq \mathbb{E}_{y_i=0}\big[Z_i \mathbf{1}\{\mathcal{G}_\delta(x_i)\}\big] + C_{\mathrm{mv}}\, \mathbb{P}_{y=0}\big(\neg\mathcal{G}_\delta(x)\big)$$

$$\leq \mathbb{E}_{y_i=0}\big[Z_i \mathbf{1}\{\mathcal{G}_\delta(x_i)\}\big] + \frac{C_{\mathrm{mv}}}{\delta} L_{RR}. \tag{17}$$

Therefore, the total margin–violation error satisfies

$$\mathcal{E}_{\mathrm{mv}} = \mathbb{E}_{y_i=0}[Z_i] + \mathbb{E}_{y_j=1}[W_j]$$

$$\leq \Big(\mathbb{E}_{y_i=0}\big[Z_i \mathbf{1}\{\mathcal{G}_\delta(x_i)\}\big] + \mathbb{E}_{y_j=1}[W_j]\Big) + \frac{C_{\mathrm{mv}}}{\delta} L_{RR}. \tag{18}$$

By construction,

$$\mathbb{E}_{y_i=0}\big[Z_i \mathbf{1}\{\mathcal{G}_\delta(x_i)\}\big] \leq \mathbb{E}_{y_i=0}[Z_i],$$

so the bracketed term in Eq. 18 is bounded above by the full margin–violation error:

$$\mathbb{E}_{y_i=0}\big[Z_i \mathbf{1}\{\mathcal{G}_\delta(x_i)\}\big] + \mathbb{E}_{y_j=1}[W_j] \leq \mathcal{E}_{\mathrm{mv}}. \tag{19}$$

Proposition 1 states that for the flow $\Phi_{\theta^*}$ minimizing $L_{\text{ML}} + \lambda_1 L_{\text{BO}}$ we have

$$\mathcal{E}_{\text{mv}} \ \leq \ (B_n - B_a)\, L_{\text{BO}}(\Phi_{\theta^*}) + \frac{N}{N+M}\, C_B. \tag{20}$$

Since the bracketed term in Eq. 18 is at most $\mathcal{E}_{\text{mv}}$, Eq. 20 implies

$$\mathbb{E}_{y_i=0}\big[Z_i\,\mathbf{1}\{\mathcal{G}_\delta(x_i)\}\big] + \mathbb{E}_{y_j=1}[W_j] \ \leq \ (B_n - B_a)\, L_{\text{BO}}(\Phi_{\theta^*}) + \frac{N}{N+M}\, C_B.$$

Substituting this into Eq. 18 yields

$$\mathcal{E}_{\text{mv}} \ \leq \ (B_n - B_a)\, L_{\text{BO}}(\Phi_{\theta^*}) + \frac{N}{N+M}\, C_B + \frac{C_{\text{mv}}}{\delta}\, L_{RR},$$

which is exactly Eq. 15.

