# OpenReview forum: "Learning Discriminative and Generalizable Anomaly Detector for Dynamic Graph"
_ICLR.cc/2026/Conference — ICLR 2026 Conference Withdrawn Submission_

### Official Review · Reviewer_hg6A · 2025-10-30

**Soundness:** 2
**Presentation:** 2
**Contribution:** 2
**Rating:** 4
**Confidence:** 3

**Summary:**

This paper focuses on anomaly detection in dynamic graphs, where labeled anomalies are scarce and existing unsupervised or semi-supervised methods often fail to learn discriminative representations or generalize well. The authors propose SDGAD, a supervised framework that constructs residual representations capturing deviations from historical behavior, and constrains normal residuals within an interval defined by two co-centered hyperspheres to maintain consistent scaling while enlarging separation from anomalies. Additionally, a normalizing flow models the likelihood distribution of normal samples, yielding an explicit decision boundary and a bi-boundary optimization strategy to enhance generalization. Experiments on six datasets containing both real and synthetic anomalies demonstrate improvements over a diverse range of baselines across multiple metrics.

**Strengths:**

1. well written and no obvious typo

2. This paper proposes a new method called SDGAD to achieve anomaly detection with the labels.

**Weaknesses:**

1. I'm confused about the motivation of why focus on supervised anomaly detection rather than the unsupervised setting. In my opinion, unsupervised setting more consistent with real-world scenario.

2. The comparison in the main experiment seem to be unfair, since no anomaly detection are introduced.

3. Density estimation seems to be a time-consuming process, no matter what kind of approximate method is used. This raise the concern of scalablity.

**Questions:**

see Weaknesses

---

> ### Author Response · Authors · 2025-11-20
>
> **Weakness1: I'm confused about the motivation of why focus on supervised anomaly detection rather than the unsupervised setting. In my opinion, unsupervised setting more consistent with real-world scenario.**
>
> We thank the reviewer for the question.  In real-world scenarios, labeled anomalies are extremely limited or even unavailable. Unsupervised methods are indeed the only viable option when labeled anomalies are completely unavailable. However, studying how to make effective use of extremely limited labels is also a realistic and important problem. As highlighted in [1] (page 17) and [2] (page 19), our focus is fully aligned with the promising research direction "the open-set supervised GAD" and "imbalanced Graph Anomaly Detection", where models are trained with a very limited (0.1\%~1\%) number of labeled anomalies while assuming the presence of unseen anomaly types. This setting is more challenging and far less explored than the unsupervised one. Our work fills this gap and demonstrates that even extremely limited anomaly supervision can provide substantial performance gains while maintaining generalization ability.
>
> We also kindly refer you to our responses to Reviewer QCW6, who raised a series of closely related questions, where we provide a more complete and systematic discussion that may help clarify your concern.
>
> [1] H. Qiao et al., "Deep Graph Anomaly Detection: A Survey and New Perspectives,"  IEEE TKDE, 2025.
>
> [2] X. Ma et al., "A Comprehensive Survey on Graph Anomaly Detection With Deep Learning," IEEE TKDE,  2022
>
> **Weakness2: The comparison in the main experiment seem to be unfair, since no anomaly detection are introduced**
>
> We thank the reviewer for the comment. We are not fully certain what is meant by “no anomaly detection is introduced.”  If the reviewer is referring to the use of baselines in the main experiments that were not originally proposed for anomaly detection task, we would like to offer the following clarification.
>
>
> Our experiments include both commonly used discrete-time DGAD baselines and the most recent continuous-time DGAD methods [1,2]. Because continuous-time DGAD is still an emerging area with limited domain-specific models, we follow the standard practice in prior work [1,2,3] by also including representative continuous-time dynamic graph models originally proposed for link prediction but directly adaptable to anomaly detection, as detailed in Appendix B.2. This evaluation setup aligns with established protocols in the DGAD literatures [1,2,3], and all baselines are selected and implemented accordingly, ensuring the rigor and consistency of our experimental design.
>
> If we have misunderstood your point, we would appreciate any further clarification so that we can provide a more accurate response.
>
> [1] Y. Xiao et.al "A generalizable anomaly detection method in dynamic graphs", AAAI, 2025
>
> [2] S. Tian et.al, "SAD: semi-supervised anomaly detection on dynamic graphs", IJCAI, 2023
>
> [3] T. Poštuvan et.al, "Learning-Based Link Anomaly Detection in Continuous-Time Dynamic Graphs", TMLR, 2024 .
>
>
> **Weakness3: Density estimation seems to be a time-consuming process, no matter what kind of approximate method is used. This raise the concern of scalablity.**
>
> We thank the reviewer for the comment. In SDGAD, density estimation operates on the low-dimensional event embeddings produced by the CTDG encoder rather than on the full dynamic graph, which ensures that both training and inference **scale linearly** with the number of events. Moreover, the normalizing flow used in our framework compute exact likelihoods in closed form without iterative optimization or sampling, and their inference cost is comparable to that of a standard feed-forward network. In contrast, many unsupervised DGAD methods rely on negative sampling, contrastive objectives, and other auxiliary components that require multiple additional forward and backward passes, resulting in a higher effective training cost than our framework.
>
> In addition, we provide empirical results comparing SDGAD (with TCL as the encoder) and the unsupervised TCL-based DGAD paradigm on the MOOC dataset. The average per-epoch training and inference times are 35 s and 13 s for SDGAD, and 40 s and 9 s for the unsupervised TCL model. These results confirm that our framework maintains comparable efficiency in practice.

---

### Official Review · Reviewer_qcw6 · 2025-10-30

**Soundness:** 2
**Presentation:** 1
**Contribution:** 2
**Rating:** 2
**Confidence:** 4

**Summary:**

The paper proposes SDGAD, a supervised anomaly detection method for continuous-time dynamic graphs. Although technically sound, the motivation and contribution are questionable.

**Strengths:**

* The model achieves slightly better F1 scores than some baselines.

**Weaknesses:**

1. **Questionable motivation (Lines 11–16).**
   The motivation presented in the paper is not convincing. The authors claim that label scarcity motivates now works, but then argue that the prevalence of unsupervised or semi-supervised methods is a problem that needs to be fixed by introducing a fully supervised method. This reasoning is conceptually inconsistent. If labels are scarce, it is perfectly reasonable for the community to focus on unsupervised or semi-supervised settings. Turning to a supervised setting does not address the stated problem; it avoids it.

2. **Supervised setting reduces to standard binary classification.**
   Once anomaly labels are available, the task essentially becomes a binary classification problem (albeit with imbalanced data). In that case, numerous well-established methods exist for handling class imbalance (e.g., weighted loss, focal loss, re-sampling, or cost-sensitive learning). The proposed method offers no clear advantage over these simpler, well-understood approaches, and the authors do not provide a convincing justification for introducing a more complex framework.

3.  **Misalignment with the anomaly detection community’s focus.**
   The current trend and practical relevance of anomaly detection lie in *unsupervised* or *semi-supervised* learning, since real-world anomalies are diverse and often lack explicit labels. Relying on labeled anomalies restricts the model to only those anomaly types seen during training, severely limiting generalization and defeating the purpose of anomaly detection. The proposed approach thus misaligns with the core philosophy and goals of the field.

4. **Limited novelty and weak experimental rigor.**
   The proposed components are rather standard and appear to be a straightforward combination of existing techniques (residual representation, boundary-based loss, and normalizing flow). There is little conceptual or algorithmic innovation. Moreover, several baselines compared in the experiments are outdated or not representative of recent state-of-the-art anomaly detection methods, which undermines the fairness and credibility of the evaluation. Additionally, the relatively newer baselines are not methods in this field.

**Questions:**

See weaknesss.

---

> ### Author Response · Authors · 2025-11-20
>
> **Weakness1: "Questionable motivation..."**
>
> Thanks for your comment. The concern appears to stem from the concise wording in the abstract(Lines 11–16). We would like to clarify our motivation more explicitly as state in the introduction (lines 39-70):
>
> Unsupervised methods have become dominant primarily because the scarcity of available labeled anomalies, leading researchers to design methods that do not rely on labeled anomalies.  However, the absence of explicit supervision often causes the model to learn weakly discriminative representations and produce ambiguous decision boundaries. These limitations cannot be fundamentally resolved by making the unsupervised objectives more complex.
>
> While available labeled anomalies are indeed scarce, they are not completely absent. Some semi-supervised methods attempt to leverage the few available labeled anomalies, but they often struggle to generalize to unseen anomalies due to the limited supervision and biased pseudo-labeling.
>
> Thus, how to effectively utilizing the scarce but available anomaly labels remains an important and underexplored problem. We do not avoid the label sparsity issue. Our motivation to study how to learn a model that achieves both strong discriminative ability and good generalization when trained with such extremely limited anomaly labels.
>
> **Weakness2: "Supervised setting reduces to standard binary classification..."**
>
> Although supervised anomaly detection on general data (e.g., images) can be regarded as an imbalanced binary classification problem and can be address by some classical imbalance learning methods, this formulation is overly simplistic and inapplicable in the context of dynamic graph anomaly detection.
>
> First, standard imbalance learning methods (e.g., re-weighting, re-sampling, and cost-sensitive learning) rely on the assumption that samples are independent and identically distributed (i.i.d), which is fundamentally violated in dynamic graphs where samples are typically non-i.i.d.
>
> Second, these methods assume that anomalies follow a single stationary distribution, while anomalies in dynamic graphs are heterogeneous, evolve over time, and often include new types that do not appear during training. As a result, treating all anomalies in a dynamic graph as a fixed class and applying standard imbalance learning strategies fails to capture their temporal and structural variability and does not address the core challenges of this setting.
>
> Even though the above limitations already indicate that classical imbalance methods are not appropriate for dynamic graph anomaly detection, we further performed additional experiments on Wikipedia dataset to validate this point. Specifically, we applied several representative re-weighting losses, including weighted cross-entropy (WCE), class-balanced loss (CBL) and focal loss (FL). These methods not only failed to improve performance but led to noticeable degradation, primarily because they easily overfit the extremely limited anomaly labels available in dynamic graphs. The detailed results are provided below.
>
>  | Method   | AUROC           | AP          | F1           |
> |----------|------------------|------------------|------------------|
> | TCL | 77.69±0.74 | 5.38±1.21  |  2.41±0.0 |
> | TCL + WCE     | 73.83 ± 1.92     | 3.74 ± 1.58      | 0.48 ± 0.30      |
> | TCL + CBL     | 66.47 ± 2.76     | 1.26 ± 1.67      | 0.07 ± 0.22      |
> | TCL + FL      | 71.12 ± 1.37     | 2.91 ± 1.42      | 1.12 ± 0.27      |

---

> ### Author Response · Authors · 2025-11-20
>
> **Weakness3: "Misalignment with the anomaly detection community’s focus..."**
>
> We appreciate the reviewer’s feedback, but we respectfully disagree with the reviewer’s claim that our approach misaligns the core philosophy and goals of the anomaly detection community.
>
> To clarify our standpoint, we provide concrete evidence from recent literature. As discussed in [1] (page 17) and [2] (page 19), our setting is fully aligned with the research perspective "the open-set supervised GAD" and "imbalanced Graph Anomaly Detection" where models are trained with a very limited (0.1\%~1\%) number of labeled anomalies while assuming the presence of unseen anomaly types. How to efficiently use such scarce labels is a realistic and important setting, and it is more challenging and less explored than the purely unsupervised case. Therefore, it is not reasonable to claim that our method conflicts with the goals of the community simply just because it does not target the unsupervised setting. Different settings address different practical needs, and studying how to make effective use of such limited anomaly labels is both meaningful and necessary. Our work fills this gap and demonstrates that even such scarce anomaly supervision can substantially improve performance and maintain generalization ability.
>
> The reviewer also claims that “using labeled anomalies limits generalization to unseen anomaly types,” but this challenge is exactly what we aim to address in this paper (lines 49-53). Our residual restriction (lines 242-244) and bi-boundary optimization prevent the model from relying on the specific types of the few seen anomalies, allowing it to generalize beyond them. This directly addresses the reviewer’s concern rather than suffering from it.
>
>
> [1] H. Qiao et al., "Deep Graph Anomaly Detection: A Survey and New Perspectives,"  IEEE TKDE, 2025.
>
> [2] X. Ma et al., "A Comprehensive Survey on Graph Anomaly Detection With Deep Learning," IEEE TKDE,  2022
>
> **Weakness4:"Limited novelty and weak experimental rigor..."**
>
> We respectfully disagree with the reviewer’s assessment regarding both the novelty and the experimental rigor of our work.
>
> Novelty: Although concepts such as residual representation, boundary-based loss have been explored in other domains (e.g. image anomaly detection), none of them has been defined, adapted, or implemented for dynamic graph anomaly detection. The components in our framework are introduced to solve domain-specific problems in dynamic graph anomaly detection and are designed to work jointly in a tightly coupled manner. They cannot be reproduced by any straightforward combination or minor modification of existing methods from other domains.
>
> For example, in image anomaly detection, the residual is often defined as the difference between a test image and a representative normal template (for instance, the mean of normal samples) or its reconstruction by a model trained only on normal data. This definition relies on a fixed Euclidean grid and a globally aligned notion of “normal appearance,” and it cannot be transferred to dynamic graphs, where anomalies arise from irregular interactions over an evolving topology rather than from per-instance deviations in pixel intensity. In our setting, there is no meaningful “average normal event” to subtract. Instead, we define the residual based on the characteristics of anomalies in dynamic graphs, by contrasting node embeddings with and without the current interaction so that the residual explicitly measures deviation from historical temporal–structural patterns. Thus, although the term “residual” is shared, its mathematical form and the anomaly signal it captures are fundamentally different and tailored to our domain.
>
> Experimental rigor:  Our experiments include both commonly used discrete-time DGAD baselines and the most recent continuous-time DGAD methods [1,2]. Because continuous-time DGAD is still an emerging area with limited domain-specific models, we follow the standard practice in prior work [1,2,3] by also including representative continuous-time dynamic graph models originally proposed for link prediction but directly adaptable to anomaly detection, as detailed in Appendix B.2. This evaluation setup aligns with established protocols in the DGAD literatures [1,2,3], and all baselines are selected and implemented accordingly, ensuring the rigor and consistency of our experimental design.
>
>
> [1] Y. Xiao et.al "A generalizable anomaly detection method in dynamic graphs", AAAI, 2025
>
> [2] S. Tian et.al, "SAD: semi-supervised anomaly detection on dynamic graphs", IJCAI, 2023
>
> [3] T. Poštuvan et.al, "Learning-Based Link Anomaly Detection in Continuous-Time Dynamic Graphs", TMLR, 2024 .

---

### Official Review · Reviewer_xTeB · 2025-11-01

**Soundness:** 2
**Presentation:** 2
**Contribution:** 2
**Rating:** 4
**Confidence:** 4

**Summary:**

This paper proposes SDGAD, a supervised framework for dynamic graph anomaly detection that combines three components: (1) residual representation encoding that captures deviations from historical patterns, (2) representation restriction using co-centered hyperspheres, and (3) bi-boundary optimization with normalizing flows.

**Strengths:**

1. The paper effectively articulates the limitations of unsupervised and semi-supervised approaches in DGAD, particularly the issue of ambiguous decision boundaries.

2. The evaluation spans six datasets with both real and synthetic anomalies, comparing against 15 baseline methods across multiple metrics (AUROC, AP, F1).

3. The framework can be integrated with different CTDG encoders (TCL, CAWN, DyGFormer), demonstrating some generalizability.

**Weaknesses:**

1. The representation restriction mechanism (Section 4.2, Equation 5) is directly borrowed from Zhang et al. (2024) "Deep orthogonal hypersphere compression for anomaly detection." The authors acknowledge drawing "inspiration" but the formulation is nearly identical—using co-centered hyperspheres with interval penalties. The only modification is adding MSE and cosine similarity terms for anomalies, which is a minor incremental change.

2. Using normalizing flows to model normal distributions and detect anomalies as out-of-distribution samples is standard practice (Kirichenko et al., 2020; Kumar et al., 2021, both cited). The bi-boundary optimization (Section 4.3) essentially applies a margin to the likelihood threshold, which is conceptually similar to margin-based losses in classification (e.g., hinge loss, triplet loss).


3. The framework requires careful tuning of multiple hyperparameters (λ₁, λ₂, rₘᵢₙ, rₘₐₓ, α, τ, L) across different components. Tables 6-8 show high sensitivity to these choices, suggesting the method lacks robustness. The paper admits "there is no configuration simultaneously maximizes all metrics" (page 16, line 825)

4. The paper doesn't compare against recent anomaly detection methods using normalizing flows (e.g., "Inflow: Robust outlier detection utilizing normalizing flows" by Kumar et al., 2021, which is cited but not compared).

**Questions:**

1. Can you provide rigorous justification for why residual representations are theoretically optimal for dynamic graph anomalies?

2. How does your hypersphere restriction differ substantively from Zhang et al. (2024) beyond the minor additions in Equation 5?

3. Can you provide tighter theoretical bounds that account for all three loss terms and their interactions?

---

> ### Author Response · Authors · 2025-11-20
>
> **Weakness1&Question2: "...the formulation is nearly identical..." & "How does your hypersphere restriction
> differ substantively from Zhang et al. (2024) beyond the minor additions in Equation 5"**
>
> We appreciate the reviewer’s comparison with DO2HSC[1]. Although our representation restriction mechanism draws on the intuition of an interval constraint, its optimization process is fundamentally distinct.
>
> In DO2HSC, the radii $r_{\min}$ and $r_{\max}$ are not hyperparameters. They are determined through a mandatory warm-up stage: DOHSC (the single-hypersphere variant of DO2HSC) is first trained to obtain a stable center $c$ and distance distribution ${d_i}$, after which empirical percentiles are used to fix the two radii. The bi-hypersphere loss is then optimized with these radii frozen, resulting in a multi-stage pipeline.
>
> In contrast, our restriction operates directly on a smooth norm of the projected residual representation and does not depend on any learned center or empirical statistics. In our formulation, $r_{\min}$ and $r_{\max}$ are ordinary scalar hyperparameters trained jointly with the rest of the model. No warm-up, no percentile estimation, and no auxiliary hypersphere fitting are required.
>
>
> [1] Zhang et al., "Deep orthogonal hypersphere compression for anomaly detection," NeurIPS, 2024.
>
>
> **Weakness2&Weakness4: Using normalizing flows to model normal distributions and detect anomalies as out-of-distribution samples is standard practice. The bi-boundary optimization (Section 4.3) essentially applies a margin to the likelihood threshold, which is conceptually similar to margin-based losses in classification (e.g., hinge loss, triplet loss).  The paper doesn't compare against recent anomaly detection methods using normalizing flows (e.g., "Inflow: Robust outlier detection utilizing normalizing flows" by Kumar et al., 2021, which is cited but not compared)**
>
> We appreciate the reviewer’s feedback.
>
> First, flow-based anomaly detection methods such as InFlow operate on independent tabular or vector inputs, and applying them to dynamic graphs would require adding a temporal graph encoder, making it impossible to separate the encoder’s effect from that of the flow and thus unsuitable as a fair baseline.
>
> Moreover, in our framework, the normalizing flow serves only as a density estimator on the learned residual representations and outputs samples' log-likelihoods that are treated as anomaly scores. Our main contribution lies in overall dynamic graph anomaly detection framework, including residual representation encoding, representation restriction, and the bi-boundary optimization, rather than the choice of a specific density estimator architecture. We cite InFlow to highlight the typical-set issue of flow-based models and to motivate the design of our bi-boundary optimization.
>
>
> The margin in bi-boundary optimization serves a fundamentally different purpose from classical margin-based losses, and the two cannot be treated as equivalent. The margin in margin-based losses is directly imposed on class scores or distances to enlarge the separation between known classes. As a result, it ties the decision boundary to the specific anomaly types seen during training, which limits generalization to unseen anomalies. In contrast, our margin is defined within the likelihood space of a density estimator trained only on normal samples. Its purpose is to form a buffer zone that captures deviations from the likelihood distribution of normal data and labeled anomalies are used only to guide the placement of this buffer. Consequently, the decision boundary is shaped by the likelihood distribution pattern of normal samples rather than by the characteristics of the few labeled anomalies, which supports better generalization to unseen anomaly types.

---

> ### Author Response · Authors · 2025-11-20
>
> **Weakness3: The framework requires careful tuning of multiple hyperparameters across different components. Tables 6-8 show high sensitivity to these choices, suggesting the method lacks robustness. The paper admits "there is no configuration simultaneously maximizes all metrics" (page 16, line 825)**
>
> Thank you for your comment. We would like to first clarify the notion of **robustness**: a method is typically considered non-robust only when small perturbations to hyperparameters lead to abrupt, erratic, or unpredictable degradation of performance. In contrast, the hyperparameter studies across all our experiments do not exhibit such behavior. Instead, the results show **smooth and consistent trends** that align with the intended functional roles of each component, rather than any sign of instability.
>
> Regarding the statement “there is no configuration that simultaneously maximizes all metrics,” our intention was to point out the inherent property of evaluation metrics. AUROC evaluates the overall ranking of scores between normal and anomalous samples and does not depend on a specific threshold. F1 and AP, instead, depend strongly on how the model behaves near one threshold and on how closely it matches a very small set of labeled anomalies under heavy class imbalance. Thus, tuning hyperparameters to improve AUROC tends to smooth the score distribution and stabilize the global ranking, but this threshold choice is usually not the one that gives the best F1/AP. Conversely, tuning to maximize F1/AP on few positives often makes the decision boundary fit them very tightly, which can harm the overall ranking and reduce AUROC.
>
> Therefore, no single configuration can make all three metrics peak at the same time, and this divergence is a structural property of the metrics rather than evidence of instability in our method.
>
> **Question1 :Can you provide rigorous justification for why residual representations are theoretically optimal for dynamic graph anomalies?**
>
> Thanks for your question. We would like to clarify that the paper does not claim, and is not intended to imply that residual representations are theoretically optimal for dynamic graph anomalies in a strict theoretical sense. Instead, the residual representation is a conceptually motivated design choice, consistent with the common view in dynamic graph anomaly detection that anomalies are events deviating from the expected behaviour of the system conditioned on its history.
>
> **Question3: Can you provide tighter theoretical bounds that account for all three loss terms and their interactions?**
>
> Thanks for your suggestion. Our error analysis in Appendix D.2 just focuses on two loss terms ($L_{ML}$ and $L_{BO}$) because both of them belong to the normalizing flow module. Proposition 1 therefore shows that, for fixed representations, how the margin–violation error is controlled by the boundary loss. In contrast, $L_{RR}$ is optimized in a separate module and does not directly influence the likelihood or the bi–boundary mechanism analyzed in Proposition 1.
>
> To address the reviewer’s request for a bound involving all three losses, we have added an extended error analysis (now Appendix D.3 in the revised paper).

---

### Official Review · Reviewer_5TWh · 2025-11-01

**Soundness:** 2
**Presentation:** 3
**Contribution:** 1
**Rating:** 4
**Confidence:** 3

**Summary:**

This paper proposes SDGAD, a supervised framework for dynamic graph anomaly detection (DGAD), aiming to improve discriminability in unsupervised settings and enhance generalization to unseen anomalies. The framework consists of three main components. First, Residual Representation Encoding calculates the difference between node embeddings with and without the current interaction, thus highlighting deviations from historical patterns. Second, Representation Restriction constrains the residuals of normal samples within an interval bounded by two co-centered hyperspheres, which helps compact their scale while keeping anomalies separable. Third, Bi-Boundary Optimization with Normalizing Flows models the likelihood distribution of normal samples and introduces explicit normal and anomaly boundaries separated by a margin to improve robustness.

**Strengths:**

1. The paper is clearly written, with a clear problem statement and a logically organized presentation.
2. SDGAD improves both ranking metrics (AUROC/AP) and classification metrics (F1) across multiple datasets and backbones.

**Weaknesses:**

1. Although SDGAD demonstrates strong performance in supervised settings, it requires a sufficient number of labeled anomalies to train effectively. In many practical applications, such as fraud detection, labeled anomalies are extremely rare or unavailable, which could reduce the feasibility and impact of the proposed approach in such environments.
2. The complexity of the loss design, which combines residual restriction and bi-boundary optimization, adds tuning complexity ($\lambda_{1}$, $\lambda_{2}$, $\tau$, $r_{\mathrm{min}}$, $r_{\mathrm{max}}$, $\alpha$), though the authors provide sensitivity analysis.

**Questions:**

1. How does the proposed dynamic graph anomaly detection task fundamentally differ from a standard imbalanced classification problem in terms of definition, modeling, and challenges?
2. How does SDGAD perform when anomaly labels are extremely limited (e.g., < 0.1%)?
3. Have other density estimators (e.g., energy-based models, autoregressive likelihood models) been tested instead of normalizing flows? Would they face the same “typical set” risk?

---

> ### Author Response · Authors · 2025-11-20
>
> **Weakness 1: "...reduce the feasibility and impact of the proposed approach in such environments."**
>
>
> We thank the reviewer for the comment.  However, we respectfully disagree with the reviewer's statement that SDGAD requires a sufficient number of labeled anomalies. As shown in Table.5 and Appendix A.2, all real-world datasets contain very limited (0.1\%~1\%)  labeled anomalies, and our model is trained under exactly this setting. This aligns with the reviewer’s claim: in many practical applications such as fraud detection, labeled anomalies are **extremely rare**.
>
> We agree that when anomaly labels are completely unavailable, unsupervised methods are the only viable option. However, different settings address different practical needs, and learning how to make effective use of extremely rare anomaly labels is also a realistic and important problem.
>
> As highlighted in recent literature [1] (page 17) and [2] (page 19), our focus is fully aligned with the promising research direction "the open-set supervised GAD" and "imbalanced Graph Anomaly Detection", where models are trained with a very limited (0.1\%~1\%) number of labeled anomalies while assuming the presence of unseen anomaly types.  It is more challenging and less explored than the unsupervised setting. Our work fills this gap and demonstrates that even extremely limited anomaly supervision can substantially improve performance and maintain generalization ability.
>
> [1] H. Qiao et al., "Deep Graph Anomaly Detection: A Survey and New Perspectives,"  IEEE TKDE, 2025.
>
> [2] X. Ma et al., "A Comprehensive Survey on Graph Anomaly Detection With Deep Learning," IEEE TKDE,  2022
>
> **Weakness 2: "... tuning complexity..."**
>
> Thanks for your comment.  In discussions of tuning complexity, a method is typically considered difficult to tune only when small hyperparameter changes lead to unstable or unpredictable performance. This is not the case for our method. As shown in our hyperparameter studies, varying each hyperparameter produces smooth and consistent trends, and the effect of each component matches its intended role in the framework. In practice, most hyperparameters do not need heavy dataset-specific retuning; for example, we can use $\tau = 0.1$  and $\alpha = 0.01$ across all experiments.
>
> Our sensitivity analysis also shows the  inherent property of the evaluation metrics in anomaly detection. AUROC evaluates the global ranking between normal and anomalous samples and is independent of any particular threshold. In contrast, F1 and AP depend strongly on the model behavior around a single decision threshold and on how well it captures a very small set of labeled anomalies under severe class imbalance. Consequently, tuning hyperparameters to maximize AUROC tends to smooth the score distribution and stabilize the overall ranking, which is not necessarily the configuration that yields the best F1/AP. Conversely, tuning for F1/AP on few positives often makes the decision boundary fit these labeled anomalies very tightly, which can slightly harm the global ranking and reduce AUROC.
>
> Therefore, no single hyperparameter configuration can make AUROC, F1, and AP peak simultaneously. This divergence is a structural property of the metrics under class imbalance, not evidence that our method is unstable or excessively difficult to tune.

---

> ### Author Response · Authors · 2025-11-20
>
> **Question1:"...imbalanced classification problem..."**
>
>  Although supervised anomaly detection on general data (e.g., images) can be regarded as an imbalanced binary classification problem and can be address by some classical imbalance learning methods, this formulation is overly simplistic and inapplicable in the context of dynamic graph anomaly detection.
>
> First, standard imbalance learning methods (e.g., re-weighting, re-sampling, and cost-sensitive learning) rely on the assumption that samples are independent and identically distributed (i.i.d), which is fundamentally violated in dynamic graphs where samples are typically non-i.i.d.
>
> Second, these methods assume that anomalies follow a single stationary distribution, while anomalies in dynamic graphs are heterogeneous, evolve over time, and often include new types that do not appear during training. As a result, treating all anomalies in a dynamic graph as a fixed class and applying standard imbalance learning strategies fails to capture their temporal and structural variability and does not address the core challenges of this setting.
>
> We further performed additional experiments on Wikipedia dataset to validate this point. Specifically, we applied several representative imbalance-learning losses, including weighted cross-entropy (WCE), class-balanced loss (CBL) and focal loss (FL). These methods not only failed to improve performance but led to noticeable degradation, primarily because they easily overfit the extremely limited anomaly labels available in dynamic graphs. The detailed results are provided below.
>
>  | Method   | AUROC           | AP          | F1           |
>  |----------|------------------|------------------|------------------|
> | TCL | 77.69±0.74 | 5.38±1.21  |  2.41±0.0 |
>  | TCL + WCE     | 73.83 ± 1.92     | 3.74 ± 1.58      | 0.48 ± 0.30      |
>  | TCL + CBL     | 66.47 ± 2.76     | 1.26 ± 1.67      | 0.07 ± 0.22      |
>  | TCL + FL      | 71.12 ± 1.37     | 2.91 ± 1.42      | 1.12 ± 0.27      |
>
> **Question2:"How does SDGAD perform when anomaly labels are extremely limited (e.g., < 0.1\%)?"**
>
> Thanks for your question. This setting is already covered in our experiments. As stated in lines 691–692 "On the Reddit dataset, anomalies are extremely sparse, with anomaly ratios of 0.024\% in the training set, 0.065\% in the validation set, and 0.08\% in the test set." SDGAD still achieves clear gains over all baselines across all metrics on this dataset.
>
> **Question3: "Have other density estimators (e.g., energy-based models, autoregressive likelihood models) been tested instead of normalizing flows? Would they face the same “typical set” risk?"**
>
>
> The “typical set” phenomenon arises because, in high-dimensional distributions, most probability mass lies in a narrow region where samples share similar average log-likelihood, rather than at points with the maximum likelihood. As a result, likelihood-based models [1] such as normalizing flows and autoregressive models may assign high likelihood to inputs that fall within this statistically common region even when they are semantically atypical. Energy-based models show a similar effect [2].
>
> Although this phenomenon affects all density estimators, we use normalizing flows because they provide an explicit and tractable density via an invertible transformation, allowing exact likelihood computation and consistent use of the learned latent representation. Their bidirectional mapping avoids the ordering constraints and sequential computation required by autoregressive models, and their training does not involve estimating an intractable partition function as in energy-based models.
>
>
> [1] E. Nalisnick et.al., "Detecting Out-of-Distribution Inputs to Deep Generative Models Using Typicality," ,Arxiv, 2019
>
> [2] W. Liu et.al., "Energy-based Out-of-distribution Detection," , NeurIPS, 2020

---

> > ### Comment · Reviewer_5TWh · 2025-11-27
> >
> > Thanks for the reply. Most of my concerns have been addressed. I will improve my score.

---

> > > ### Author Response · Authors · 2025-11-27
> > >
> > > Dear Reviewer 5TWh,
> > >
> > > We are happy to read that you are satisfied with our rebuttal, and thank you for your support in our work.
> > >
> > > With kindest regards,
> > >
> > > Authors.

---

### Author Response · Authors · 2025-12-03
**Summary for Area Chair and reviewers**

Dear Area Chair,

We would like to express our sincere gratitude for your time and effort in overseeing the review process.  For your convenience, we first summarize the rebuttal and author–reviewer discussion. Reviewer 5TWh acknowledged that most concerns were addressed and  increased the score to 6. Reviewers xTeB, qcw6, and hg6A did not have the opportunity to participate due to the early termination of the discussion period. Below we summarize the main reviewer concerns and our corresponding rebuttals.

**Questionable motivation**: The primary concern raised by most reviewers (5TWh, qcw6, hg6A) was the paper’s focus on the supervised rather than the unsupervised setting. Reviewer qcw6 further commented that this focus is misaligned with the goals of the anomaly detection community.

We clarified that (1) our motivation is not to show that supervised anomaly detection outperforms unsupervised approaches, but to study how to maximally leverage extremely scarce anomaly labels, so limited that most existing methods resort to unsupervised learning, while still ensuring generalization to unseen anomaly types. (2) We also provided concrete evidence from recent literature [1,2], showing that our setting aligns with promising community directions in open-set supervised GAD and imbalanced graph anomaly detection, where models are trained with extremely few labeled anomalies while assuming unseen anomaly types.


**Supervised Setting Reduces to Binary Classification**: Two reviewers (5TWh, qcw6) argued that the supervised setting may reduce to standard binary classification and could be addressed using generic imbalanced learning methods.

We argue that, although supervised anomaly detection on i.i.d. data can often be framed as imbalanced binary classification, this formulation is overly simplistic for dynamic graph anomaly detection. Standard imbalanced learning methods typically assume i.i.d. samples and a stationary anomaly distribution, both of which are violated in dynamic graphs where observations are non-i.i.d. and anomalies are heterogeneous, time-evolving, and may be unseen during training. We also provided additional experiments show that directly applying generic imbalanced learning methods can even degrade performance.


**Sensitivity to hyperparameters**: Two reviewers (5TWh, xTeB) noted that the model includes multiple hyperparameters and may be sensitive to their choices, raising robustness concerns. Reviewer xTeB also pointed out that out paper states, “there is no configuration that simultaneously maximizes all metrics.”

.
We clarified that a method is non-robust only if small perturbations in hyperparameters cause abrupt or erratic performance drops, which is not observed in our studies. Instead, performance varies smoothly and consistently with each component’s intended role. In practice, most hyperparameters do not require extensive dataset-specific tuning.

Furthermore, the divergence is an inherent property of the metrics rather than instability of our method. AUROC is threshold-free and reflects global ranking quality, whereas F1 and AP are threshold-dependent and dominated by near-boundary behavior under extreme class imbalance. Consequently, optimizing AUROC often smooths score distributions but may not yield the best F1/AP, while tuning for F1/AP can overfit the limited positives and reduce AUROC.

**Experiment rigor**: Two reviewers (5TWh, xTeB) questioned the rigor of our evaluation, arguing that several baselines are outdated and that newer ones are not representative of anomaly detection methods.

We clarified that our experiments include both commonly used discrete-time DGAD baselines and the most recent continuous-time DGAD methods [3,4]. Because continuous-time DGAD remains an emerging research area with few domain-specific models, we follow established practice in prior work [3–5] by also including representative continuous-time dynamic graph models originally proposed for link prediction but directly adaptable to anomaly detection. As detailed in Appendix B.2, this setup aligns with accepted evaluation protocols in DGAD literature [3–5] and ensures a fair and rigorous comparison.

[1] H. Qiao et al., "Deep Graph Anomaly Detection: A Survey and New Perspectives,"  IEEE TKDE, 2025.

[2] X. Ma et al., "A Comprehensive Survey on Graph Anomaly Detection With Deep Learning," IEEE TKDE,  2022

[3] Y. Xiao et.al "A generalizable anomaly detection method in dynamic graphs", AAAI, 2025

[4] S. Tian et.al, "SAD: semi-supervised anomaly detection on dynamic graphs", IJCAI, 2023

[5] T. Poštuvan et.al, "Learning-Based Link Anomaly Detection in Continuous-Time Dynamic Graphs", TMLR, 2024 .


Finally, we would once again like to thank you, our reviewers, for your in-depth reviews. We are also thankful for our Area Chairs both the original and newly assigned chair, we strongly believe that our rebuttal has properly addressed all the concerns and questions raised by the reviewers.

Authors.

---

### Note · Authors · 2026-01-26

I have read and agree with the venue's withdrawal policy on behalf of myself and my co-authors.

---

### Meta-Review · Area_Chair_jQZT · 2025-12-03

**Summary:**

The paper presents a supervised method for anomaly detection on dynamic graphs and shows improvements across several datasets.

Reviewers appreciated the clear writing and strong empirical results. However, they raised concerns about the motivation for supervised learning with scarce labels, limited novelty, baseline fairness, and the heavy use of hyperparameters. Although the authors provided clarifications during the rebuttal and one reviewer increased their score, several key issues remain only partially addressed. The method still appears close to existing approaches, and the evaluation setup requires stronger justification. So it is suggested that the authors address all these concerns to improve their paper before submit it to another venue.

**Reviewer Concerns:**

Reviewer 5TWh's concerns were addressed.

Other concerns such as motivation and evaluation need to be addressed in a revised version.

**Reviewer Scores:**

5TWh increased its score.

Other reviewers may not change their scores.

---

### Decision · Program_Chairs · 2026-01-26

Reject